# Deletion of Topoisomerase 1 in excitatory neurons causes genomic instability and early onset neurodegeneration

Giulia Fragola [1,2], Angela M. Mabb [3], Bonnie Taylor-Blake[1,2], Jesse K. Niehaus [2], William D. Chronister[4], Hanqian Mao[1,2], Jeremy M. Simon [2,5,6], Hong Yuan[7,8], Zibo Li [7,8], Michael J. McConnell[4,9,10,11] & Mark J. Zylka[1,2✉]

Topoisomerase 1 (TOP1) relieves torsional stress in DNA during transcription and facilitates the expression of long (>100 kb) genes, many of which are important for neuronal functions. To evaluate how loss of *Top1* affected neurons in vivo, we conditionally deleted (cKO) *Top1* in postmitotic excitatory neurons in the mouse cerebral cortex and hippocampus. *Top1* cKO neurons develop properly, but then show biased transcriptional downregulation of long genes, signs of DNA damage, neuroinflammation, increased poly(ADP-ribose) polymerase-1 (PARP1) activity, single-cell somatic mutations, and ultimately degeneration. Supplementation of nicotinamide adenine dinucleotide (NAD$^+$) with nicotinamide riboside partially blocked neurodegeneration, and increased the lifespan of *Top1* cKO mice by 30%. A reduction of *p53* also partially rescued cortical neuron loss. While neurodegeneration was partially rescued, behavioral decline was not prevented. These data indicate that reducing neuronal loss is not sufficient to limit behavioral decline when TOP1 function is disrupted.

[1] Department of Cell Biology and Physiology, University of North Carolina at Chapel Hill, Chapel Hill, NC 27599, USA. [2] UNC Neuroscience Center, University of North Carolina at Chapel Hill, Chapel Hill, NC 27599, USA. [3] Neuroscience Institute, Georgia State University, Atlanta, GA 30303, USA. [4] Department of Biochemistry and Molecular Genetics, University of Virginia School of Medicine, Charlottesville, VA 22908, USA. [5] Department of Genetics, University of North Carolina at Chapel Hill, Chapel Hill, NC 27599, USA. [6] Carolina Institute for Developmental Disabilities, University of North Carolina School of Medicine, Chapel Hill, NC 27599, USA. [7] Department of Radiology, University of North Carolina at Chapel Hill, Chapel Hill, NC 27599, USA. [8] Biomedical Imaging Research Center, University of North Carolina at Chapel Hill, Chapel Hill, NC 27599, USA. [9] Department of Neuroscience, University of Virginia School of Medicine, Charlottesville, VA 22908, USA. [10] Center for Brain Immunology and Glia, University of Virginia School of Medicine, Charlottesville, VA 22908, USA. [11] Center for Public Health Genomics, University of Virginia, School of Medicine, Charlottesville, VA 22908, USA. ✉email: zylka@med.unc.edu

Normal physiological processes, including oxidative stress, neuronal activation, and gene transcription, can damage DNA in neurons[1–4]. If not precisely repaired, DNA damage creates permanent lesions in the neuronal genome, referred to as somatic mutations. Recent single-cell sequencing studies report that every neuron in the mammalian brain may contain unique somatic mutations[5–7].

DNA mutations are more abundant in individuals with neurodegenerative disorders[8–12], and reviewed in ref. [13]. In fact, one common phenotype across numerous neurodegenerative disorders, including Alzheimer's disease (AD), Parkinson's disease (PD), amyotrophic lateral sclerosis, and ataxia-telangiectasia, is an abnormal accumulation of DNA damage in neurons[13–16]. DNA damage-induced neurodegeneration is frequently associated with hyperactivation of the nicotinamide adenine dinucleotide (NAD+) consuming enzyme PARP1 (poly(ADP-ribose) polymerase 1), which leads to energetic breakdown and cell death[17–20]. A greater understanding of the genes and the molecular mechanisms that maintain genomic integrity in postmitotic neurons could thus have therapeutic relevance for multiple neurodegenerative diseases.

Decades of research in yeast and mitotic mammalian cells support a role for topoisomerase 1 (TOP1) in gene transcription and maintenance of genomic integrity[21,22]. Recent studies suggest that transcriptional stress-driven DNA damage is also prevalent in postmitotic neurons and has the potential to disrupt genes that neurons depend on most[2]. We recently identified TOP1 as a key transcriptional regulator in postmitotic neurons[23,24]. TOP1 relieves DNA supercoiling generated during transcription by catalyzing a single-strand DNA cleavage[22]. During this reaction, TOP1 transiently binds to the DNA forming a cleavage complex (TOP1cc). Acute depletion of TOP1 or acute treatment with topotecan, a TOP1 inhibitor, resulted in the downregulation of long genes (>100 kb) in cultured cortical neurons[23]. TOP1cc-dependent and -independent mechanisms were implicated in these transcriptional responses[24]. Long genes, as a class, are disproportionately associated with neuronal functions[23,25], and acute inhibition of TOP1 impaired long genes associated with synaptic function in neuronal cultures[26]. Using a single-cell RNA-sequencing (RNA-seq) approach, we also found that excitatory neurons in lower cortical layers expressed long genes to a greater extent when compared to excitatory neurons in upper cortical layers in embryonic and early postnatal mice[27].

Based on these observations, we hypothesized that TOP1 might maintain genomic integrity and/or maintain transcriptional output in the central nervous system and that deletion of Top1 would have the greatest impact on lower layer excitatory cortical neurons. Here, we tested this hypothesis by conditionally deleting Top1 in postmitotic excitatory neurons in the mouse cerebral cortex and hippocampus.

## Results

**Top1 cKO mice show motor deficits and early death.** To study the requirement for TOP1 in postmitotic excitatory neurons of the cerebral cortex and hippocampus, we crossed Top1 conditional knockout mice (Top1^fl/fl)[24] with the Neurod6^Cre mouse line[28]. Neurod6 expression begins at embryonic day 11.5 and is maintained throughout maturation in postmitotic excitatory neurons of the mouse cerebral cortex and hippocampus, except for the dentate gyrus (DG), where Neurod6 is only transiently expressed postnatally[28]. Both homozygous Top1-knockout mice (Neurod6^Cre/+::Top1^fl/fl, referred to as Top1 cKO) and heterozygous (HET) Top1 KO mice (Neurod6^Cre/+::Top1^fl/+, referred to as Top1 cHET) were generated. Neurod6^+/+::Top1^fl/fl or Top1^fl/+ mice were used as a control (wild-type (WT)). To confirm Top1

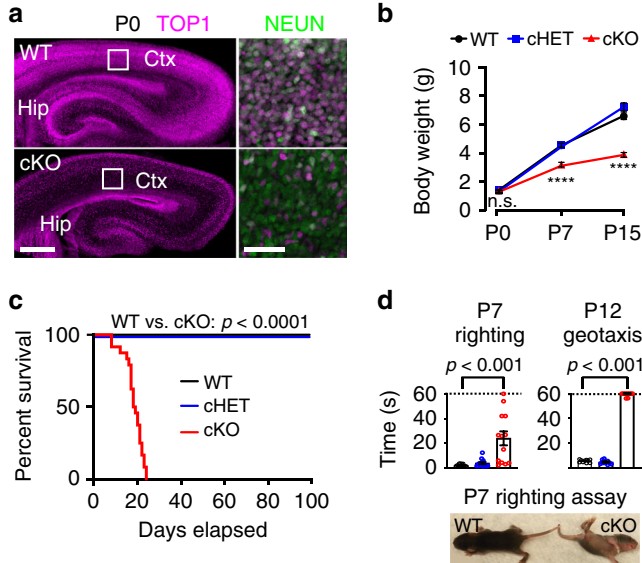

**Fig. 1 Deletion of Top1 in cortical and hippocampal neurons impairs motor function and causes premature death. a** TOP1 and (inset) NEUN immunostaining of P0 WT and Top1 cKO somatosensory cortex (Ctx) and hippocampus (Hip). Scale bar = 300 μm, inset scale bar = 50 μm. Images are representative of two independent experiments. **b** Body weight of Top1 WT cHET and cKO mice at different time points. WT: P0 $n = 10$, P7 $n = 41$, P15 $n = 37$; cHET: P0 $n = 6$, P7 $n = 27$, P15 $n = 20$; cKO: P0 $n = 6$, P7 $n = 17$, P15 $n = 12$. n.s. $p > 0.05$, **** $p < 0.0001$. WT vs. cHET $p$ values were not significant in all comparisons. One-way ANOVA with Dunnett's multiple comparison test for each time point. **c** Kaplan–Meier survival curve. Mantel–Cox test. WT vs. cHET $p > 0.05$. $n = 24$ mice per genotype. **d** Righting and geotaxis assays performed on P7 (WT = 13, cHET = 13, cKO = 13 mice) and P12 mice (WT = 10, cHET = 10, cKO = 9 mice). WT vs. cHET $p$ values were not significant ($p < 0.05$). One-way ANOVA with Dunnett's multiple comparison test. Dashed lines represent cutoff time. Values are mean and error bars are ±SEM.

deletion, we examined TOP1 protein levels at postnatal day 0 (P0) through immunostaining. TOP1 was ubiquitously expressed in WT and Top1 cHET cortex and hippocampus (Fig. 1a, Supplementary Fig. 1), but was absent in most cortical and hippocampal NEUN+ neurons of Top1 cKO mice (Fig. 1a, Supplementary Fig. 1). Cortical neurons that stained positive for TOP1 in Top1 cKO mice are most likely interneurons, which do not express Neurod6. Likewise, most of the neurons in the DG stained positive for TOP1 at this time point (Fig. 1a, Supplementary Fig. 1), consistent with transient postnatal Neurod6 expression in DG neurons.

The body weights of Top1 cKO mice were comparable to WT mice at birth, but increased to a lesser extent at P7 and P15 (Fig. 1b). Top1 cKO were viable up to the second and third postnatal week (Fig. 1c). Top1 cKO mice also showed a severe motor deficit as assessed by the righting reflex assay at P7 and the geotaxis assay at P12 (Fig. 1d). No differences in body weight, viability, or motor function were observed in Top1 cHET compared to WT mice (Fig. 1b–d). These data indicate that deletion of Top1 in postmitotic excitatory cortical and hippocampal neurons did not overtly affect brain or body development up to birth, but impaired motor function and caused premature death within the first month of life.

**Top1 cKO mice show early-onset neurodegeneration.** To determine if Top1 deletion affected postnatal brain size, we quantified brain weight in WT, Top1 cHET, and Top1 cKO mice

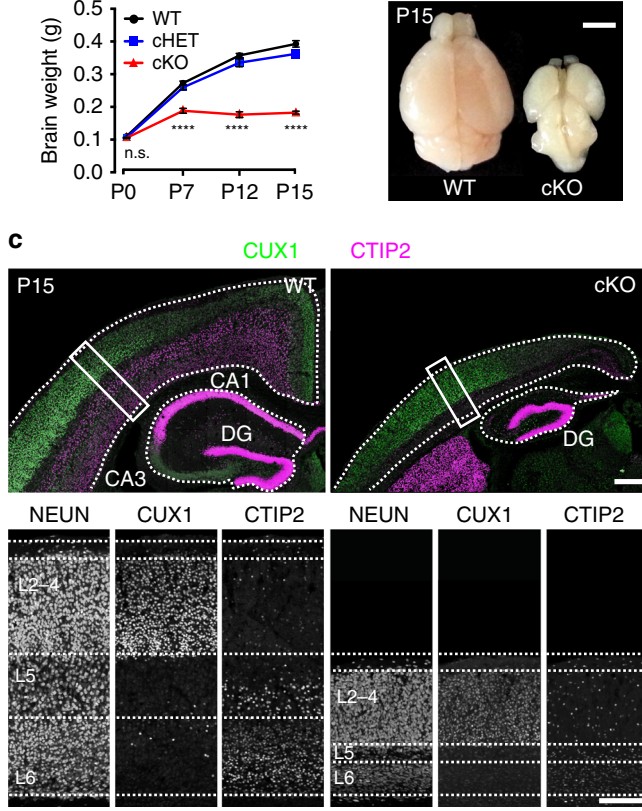

**Fig. 2 Brain, cerebral cortex, and hippocampus size reduced in *Top1* cKO mice. a** Brain weight quantification of WT (P0 *n* = 11, P7 *n* = 22, P12 *n* = 8, P15, *n* = 13), cHET (P0 *n* = 14, P7 *n* = 12, P12 *n* = 4, P15 *n* = 1) and *Top1* cKO mice (P0 *n* = 17, P7 *n* = 14, P12 *n* = 3, P15 *n* = 10). Values are mean and error bars are ±SEM. n.s. *p* > 0.05, ****p < 0.0001. WT vs. cHET *p* values were >0.05 in all comparisons. One-way ANOVA with Dunnett's multiple comparison test. **b** Representative images of P15 brains. Scale bar = 0.5 cm. **c** Sections containing the cerebral cortex and hippocampus immunostained for CUX1, CTIP2, and NEUN to identify layer 2–4, layer 5 and 6 neurons, and all neurons, respectively. Dashed lines delineate cortex, hippocampus, and cortical layers (inset). Scale bar = 300 μm, inset scale bar = 200 μm. Images are representative of three independent experiments. n.s. not significant.

at different time points. Relative to P0, WT brain weight was 2.5-fold greater at P7 and 3.5-fold greater at P15 (110.3 ± 0.003 mg at P0, 274.9 ± 0.008 mg at P7, and 381.6 ± 0.008 mg at P15) (Fig. 2a). *Top1* cKO brain weights were comparable to WT at P0, increased ~2-fold by P7, but then failed to increase at later ages (Fig. 2a), and were visibly smaller than WT brains (Fig. 2b) (101.6 ± 0.001 mg at P0, 200.2 ± 0.008 mg at P7, and 200.6 ± 0.014 mg at P15). *Top1* cHET brains were comparable to WT at all time points (106.0 ± 0.002 mg at P0, 263.6 ± 0.007 mg at P7, and 365.4 ± 0 mg at P15).

To determine if the cortex and hippocampus were affected, we immunostained P15 brain sections for NEUN (a pan-neuronal marker), CUX1 (an upper layer cortical neuron marker), and CTIP2 (enriched in lower layer cortical neurons and hippocampal granule cells) (Fig. 2c). Cortical thickness was reduced by 40% in *Top1* cKO mice relative to WT mice (Fig. 2c, Supplementary Fig. 2a). While all cortical layers were affected by *Top1* deletion, lower cortical layers showed the most prominent effect (~30% decrease in L2–4, ~70% decrease in L5, and ~40% decrease in L6) (Fig. 2c, Supplementary Fig. 2a). Hippocampal area was also drastically reduced (Supplementary Fig. 2b), with a stronger effect

in CA1, CA2, and CA3 hippocampal regions (Fig. 2c). In contrast, the DG was not overtly affected, consistent with no loss of TOP1 in this region (Supplementary Fig. 1). *Top1* cHET cortical thickness was comparable to WT (Supplementary Fig. 2a).

Given the profound reduction in cortical thickness, we next sought to understand the possible causes of this phenotype. To determine if the loss in cortical thickness was due to a deficit in cortical layering or to neurodegeneration, we compared *Top1* cKO and cHET somatosensory cortex to that of WT during the first 2 weeks of postnatal development (Fig. 3a, b). Cortical neurogenesis and neuron migration begin at E11.5, when *Neurod6^Cre* is first expressed[28], and terminate prenatally around E17.5 (reviewed in ref. [29]). At P0, the overall thickness and layer organization of *Top1* cKO and cHET cortex were comparable to that of WT, indicating that neurogenesis and neuron migration were not affected. However, *Top1* cKO cortical thickness was affected at later timepoints. In particular, L5 and L6 were more vulnerable to *Top1* loss, showing a significant decrease in thickness compared to WT at P7, and a collapse in thickness by P15. L2–4 were only significantly reduced at the P15 timepoint. *Top1* cHET cortical development was comparable to that of WT mice (Fig. 3a, b). These data suggest that homozygous loss of *Top1* caused rapid, early-onset neurodegeneration that initially affected lower layer neurons and then extended to upper layers.

CTIP2/*Bcl11b* is a transcription factor that marks projection neurons in L5 and L6 and is highly expressed in L5 corticospinal motor neurons[30]. To evaluate the extent to which the decrease in L5 thickness was due to neuronal loss over time, we quantified the density and cell size of spinal projection neurons (CTIP2^high). We found that CTIP2^high neuron density in *Top1* cKO was reduced relative to WT mice by P7, and these neurons were no longer present by P15 (Fig. 3c, d). While CTIP2^high neuron density did not decline until P7, we observed a decrease in CTIP2^high soma size beginning at P2 (Fig. 3e). These data suggest that deletion of *Top1* affected CTIP2^high L5 neurons as early as P2 and contributed to the demise of these neurons by P15. Since HET loss of *Top1* did not affect any of the neurons or markers evaluated above, we did not examine *Top1* cHET mice in subsequent experiments.

***Top1* cKO mice show neuronal apoptosis and neuroinflammation**. We next performed immunostaining for cleaved-caspase 3 (cl-CASP3) and terminal deoxynucleotidyl transferase dUTP nick-end labeling (TUNEL), markers of intermediate and late phases of apoptosis, respectively. Both markers were increased in *Top1* cKO cortex at P7, particularly in L5 and L6 where cell loss and layer thinning were most pronounced (Fig. 4a, b, Supplementary Fig. 3a, b). Another early sign of apoptosis is cell shrinkage, which was observed in L5 neurons starting at P2 (Fig. 3c, e). The cl-CASP3 signal was localized in the soma and apical dendrite of L5 neurons (Fig. 4b), suggesting that neurons in this region were undergoing apoptosis. Altogether, these data indicate that the cortex develops normally in *Top1* cKO mice up to birth, but then shows signs of neurodegeneration by P2 that initiates in lower layer excitatory neurons.

Neuroinflammation is seen in a variety of neurodegenerative diseases (reviewed in ref. [31]). To determine if neuron loss in *Top1* cKO mice was associated with astrogliosis and/or microgliosis, we immunostained P7 cortex for glial fibrillary acidic protein (GFAP) and ionized calcium-binding adaptor molecule 1 (IBA1), respectively. In contrast to WT mice, intense GFAP and IBA1 immunostaining was detected in *Top1* cKO cortex in

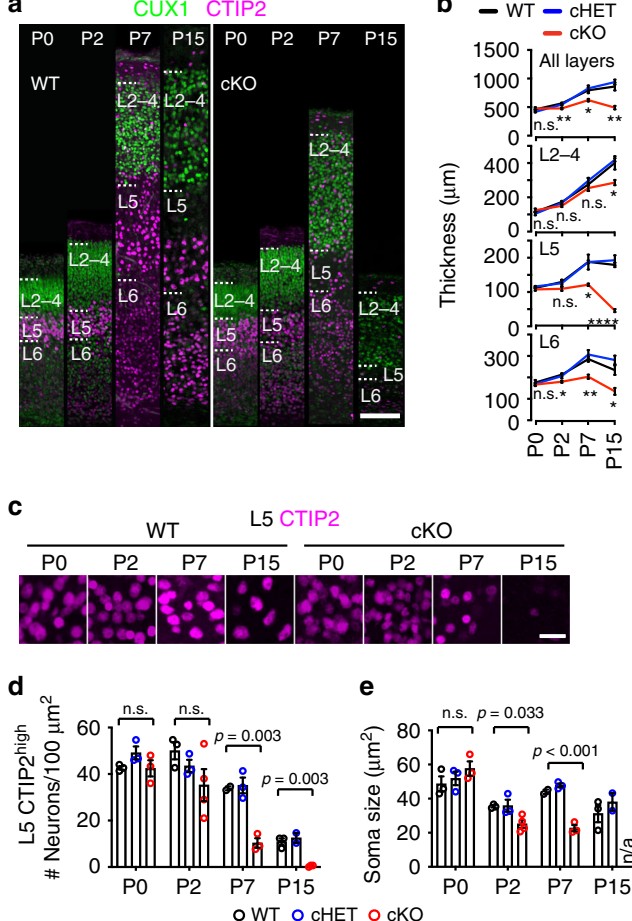

**Fig. 3 Neurodegeneration of the cerebral cortex in *Top1* cKO mice. a** WT and *Top1* cKO cerebral cortex immunostained for CUX1 and CTIP2 at P0, P2, P7, and P15. Images are representative of three independent experiments for a total of three mice per genotype per time point. Scale bar = 100 μm. **b** Cortical layer thicknesses, n = 3 per genotype per time point. n.s. p > 0.05, *p < 0.05, **p < 0.01, and ****p < 0.0001. Asterisks refer to WT vs. cKO comparisons. WT vs. cHET p values were >0.05 in all comparisons. WT vs. cKO p values in all layers: at P0 p = 0.43, at P2 p = 0.003, at P7 p = 0.029, at P15 p = 0.002. WT vs. cKO p values in L2-4: at P0 p = 0.38, at P2 p = 0.052, at P7 p = 0.57, at P15 p = 0.016. WT vs. cKO p values in L5: at P0 p = 0.28, at P2 p = 0.070, at P7 p = 0.010, at P15 p < 0.0001. WT vs. cKO p values in L6: at P0 p = 0.47, at P2 p = 0.011, at P7 p = 0.008, at P15 p = 0.012. One-way ANOVA. **c** Gradual loss of L5 CTIP2high neurons in *Top1* cKO cortex. Scale bar = 20 μm. Images are representative of three independent experiments for a total of three mice per genotype for P0, P7, and P15 and four mice per genotype for P2. **d** Quantification of L5 CTIP2high density and **e** soma size at the indicated time points. P0, P7, and P15, n = 3 for each genotype; P2, n = 3 WT, n = 4 cKO. n.s. p > 0.05. WT vs. cHET p values were >0.05 in all comparisons. n/a = not applicable because no neurons were present. One-way ANOVA with Dunnett's multiple comparison test. Values are mean and error bars are ±SEM. n.s. not significant.

L5 and L6 at P7 (Fig. 4c, d), suggesting extensive neuroinflammation in *Top1* cKO mice.

To evaluate neuroinflammation non-invasively, we performed in vivo imaging of peripheral benzodiazepine receptor (PBR) expression by positron emission tomography (PET) (Supplementary Fig. 4a, b). PBR is a mitochondrial protein found in the brain (mainly glial cells) and peripheral tissues[32]. At baseline, the brain has lower levels of PBR compared to peripheral tissues[33]. PBR

expression is increased in many neurodegenerative diseases characterized by microglia activation, including AD, PD, and multiple sclerosis[34]. PET imaging of WT and *Top1* cKO P15 mice injected with the radioisotope-labeled PBR ligand [18]F-PBR111[35] indicated a significant increase in the uptake of the ligand in the frontal lobe, cerebral cortex, and hippocampus of *Top1* cKO brains relative to WT brains (Supplementary Fig. 4a, b). Increased [18]F-PBR111 binding was also detected in sections of the cerebral cortex from *Top1* cKO mice using autoradiography (Supplementary Fig. 4a).

**Downregulation of long genes in *Top1* cKO lower layer neurons.** To evaluate the extent to which *Top1* deletion causes downregulation of long genes in vivo, we performed a small-scale single-cell RNA-seq experiment with cortical cells derived from two WT and two *Top1* cKO P7 mice. We captured a total of 1,596 cells (683 WT, 913 cKO) and identified a median of ~2,300 transcripts per cell that corresponded to a median of ~1,750 genes per cell. Using unbiased clustering analyses, we detected three main cell clusters, which we annotated as excitatory neurons, inhibitory neurons, and glial cells based on the expression of marker genes (Fig. 5a, Supplementary Data 1). We then investigated transcriptional changes between WT and cKO excitatory neurons. As expected, *Top1* expression levels were drastically decreased in excitatory neurons from cKO animals (−1.08 log2 fold change; adjusted [adj.] p value = $7.95 \times 10^{-23}$), whereas levels in inhibitory neurons (−0.43 log2 fold change; adj. p value = 0.99) and glial cells were similar between genotypes (−0.33 log2 fold change; adj. p value = 0.37) (Fig. 5b). We identified 132 downregulated genes and only one upregulated gene in excitatory neurons (Supplementary Data 2). Consistent with our previous findings[23,24], the average length of the downregulated genes in cKO excitatory neurons was 2.5-fold longer than the average gene length of all genes expressed in cortical neurons (151.4 kb vs. 59.3 kb) (Fig. 5c and Supplementary Data 2). No biased change in long gene expression was seen in the inhibitory neurons or glial cell clusters (Fig. 5c).

In the excitatory neuron cluster, many of the downregulated genes were involved in axon extension and morphology (*Celf2, Rtn4, Hspa5, Gpm6a, Zbtb18, Epha4, Gap43, Hspa5, Nefl, Ank2, Nefm,* and *Gpm6b*) and in the induction of postsynaptic currents (*Epha4, Snap25, Syt1, Nrxn1, Nrxn3, Cplx1, Snap91,* and *Snca*) (Supplementary Data 2). These data were consistent with the loss of long projecting neurons and suggested that the function of excitatory circuits in *Top1* cKO mice is affected. Genes involved in cholesterol biosynthesis and metabolic pathways (*Scd2, Sqle, Hmgcs1,* and *Hmgcr*) and genes relevant to AD (*App, Ppp3cb, Atp2a2, Calm2,* and *Scna*) were also downregulated (Supplementary Data 2). *Top1* cKO mice present a similar, but more severe, phenotype relative to mice depleted of mitogen-activated protein kinase 1/2 (MAPK1/2) in excitatory postmitotic neurons[36]. Consistent with this observation, many of the downregulated genes were involved in the MAPK signaling pathway (*Cdc42, Mef2c, Ppp3cb, Hspa5,* and *Akt3*). We also found many downregulated genes involved in calcium regulation (*Gnao1, Ywhaq, Camk4, Gnas, Atp2a2, Calm3,* and *Calr*).

Given the increased vulnerability of lower layer *Top1* cKO neurons to neurodegeneration, we asked whether long genes were more strongly affected in lower layer neurons compared to upper layer neurons. The small scale of our single-cell RNA-seq experiment precluded us from resolving upper and lower layer neuron clusters. Thus, we performed single-molecule in situ hybridization for *Ptprd* (Fig. 5d–f) (2 P7 mice per genotype, n = 11 WT sections; n = 14 cKO sections), the longest gene downregulated in cKO excitatory neurons (length 2.27 Mb; log2

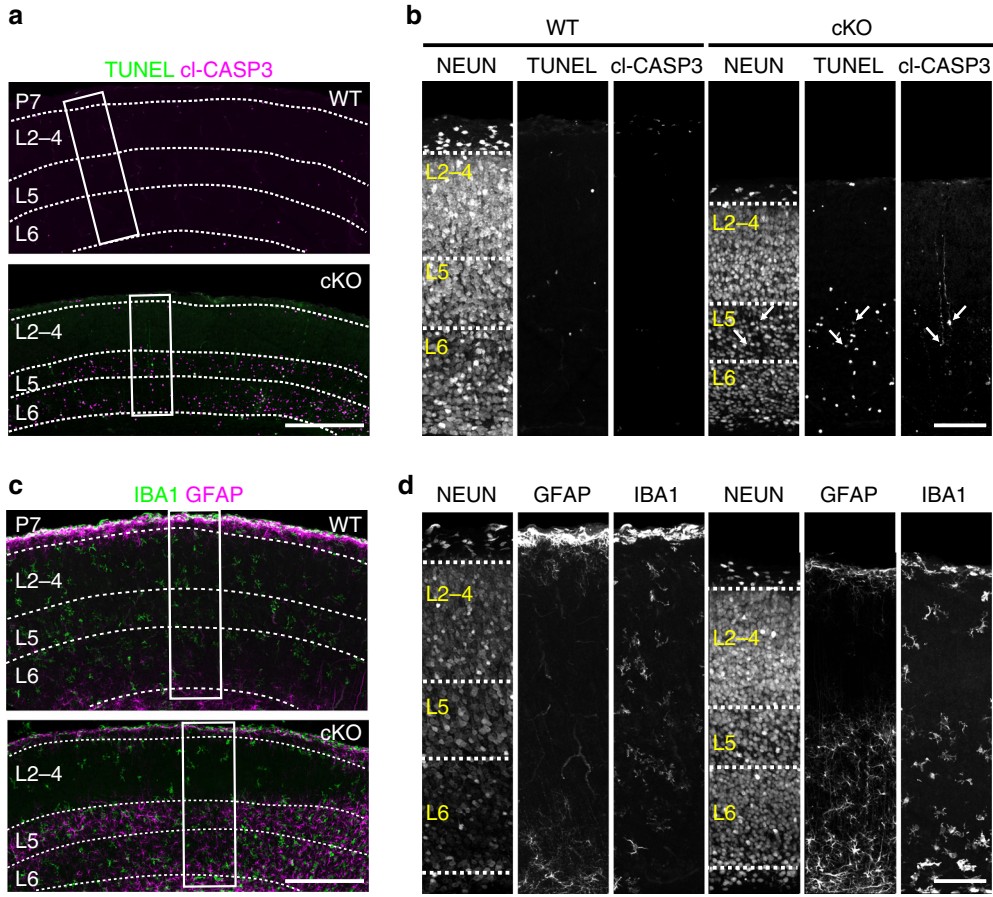

**Fig. 4 Cell death and neuroinflammation in lower cortical layers of *Top1* cKO mice. a** P7 somatosensory cortex immunostained for TUNEL, cl-CASP3, and **b** (inset) NEUN. Dashed lines delineate cortical layering identified by NEUN staining. Arrows, late apoptotic cl-CASP3⁺/TUNEL⁺ neurons show low levels of NEUN. cl-CASP3 staining localizes to the soma and apical dendrites of affected L5 neurons. Images are representative of three independent experiments for a total of three mice per genotype. **c** P7 somatosensory cortex immunostained for IBA1, GFAP, and **d** (inset) NEUN. **a**, **c** Scale bar = 300 μm, **b**, **d** scale bar = 100 μm. Images are representative of three independent experiments for a total of three mice per genotype.

fold change; −0.75; adj. $p$ value <$1.12 \times 10^{-6}$) (Supplementary Data 2). To restrict our analysis to *Neurod6*⁺ excitatory neurons, we co-hybridized the sections with a probe to *Neurod6* and used DAPI (4′,6-diamidino-2-phenylindole) to label all cells (Fig. 5d). We then used CellProfiler to quantify the number of *Ptprd* transcripts in *Neurod6*⁺ neurons (Fig. 5d–e; total number of *Neurod6*⁺ neurons identified: L2–4 WT 16,090, L2–4 cKO 18,401, L5 WT 13,929, L5 cKO 7,799, L6 WT 8,727, and L6 cKO 5,718). We observed a small decrease in *Neurod6* transcripts in L5 cKO neurons compared to WT, consistent with *Neurod6* being one of the downregulated genes in *Top1* cKO mice, and no change in L2–4 and L6 neurons (Supplementary Fig. 5a). The percentage of *Neurod6*⁺ neurons in L2–4 was similar between WT and cKO cortical sections, while in L5 and L6 the percentage of *Neurod6*⁺ neurons was lower in cKO relative to WT (Supplementary Fig. 5b), consistent with loss of lower layer excitatory neurons at P7. While the number of *Ptprd* transcripts/ *Neurod6*⁺ neuron was comparable in upper layers between WT and *Top1* cKO mice, the number of *Ptprd* transcripts/*Neurod6*⁺ neuron was significantly reduced in L5 and L6 neurons of *Top1* cKO mice (Fig. 5d–f), demonstrating reduced expression of this extremely long gene in lower, but not upper, layer neurons at P7.

**Elevated mutation burden in *Top1* cKO neurons.** TOP1 inhibition or deletion in mitotic cells induces DNA damage and genomic instability[37–40]. To evaluate the extent to which *Top1* deletion causes DNA damage in postmitotic neurons, we immunostained P0, P2, and P7 cortex for γH2AX, a marker of double-strand DNA (dsDNA) breaks. At P0 and P2 timepoints, levels of γH2AX were similar between WT and *Top1* cKO mice (Supplementary Fig. 6a). At P7, roughly one-third of neurons presented γH2AX foci in *Top1* cKO cortex. In contrast, only 3% of neurons showed γH2AX staining in P7 WT cortex (32.6 ± 5.42% neurons with γH2AX foci in *Top1* cKO vs. 3.2 ± 2.6% neurons with γH2AX foci in WT; $p$ value = 0.008, two-sided Student's $t$ test; data are the mean of percentages of neurons with γH2AX foci; $n$ = 3 mice per genotype) (Fig. 6a). γH2AX accumulation in *Top1* cKO mice extended to all cortical layers, although it was more pronounced in L6 (Fig. 6a–c, Supplementary Fig. 6b). Increased DNA damage in *Top1* cKO cortex was further confirmed by immunostaining with phosphorylated p53-binding protein 1 (Supplementary Fig. 6c, d), another marker of dsDNA breaks[41].

Extensive DNA damage, if not properly repaired, can increase somatic mutation burden. To evaluate copy number variations (CNVs) at the single-cell level, we isolated a small number of neurons ($n$ = 39 WT; $n$ = 43 cKO) by fluorescence-activated cell sorting (FACS) from P7 cerebral cortex of one WT and one *Top1* cKO mouse (Supplementary Fig. 10). We identified an increase in the number of cKO neurons with CNVs compared to WT neurons (4/39 in WT; 14/43 in cKO) (Supplementary Fig. 6e). We repeated the experiment using a droplet-based whole-genome amplification (WGA) approach on larger groups of neurons from a different pair of mice[42]. A two-fold increase in CNV prevalence

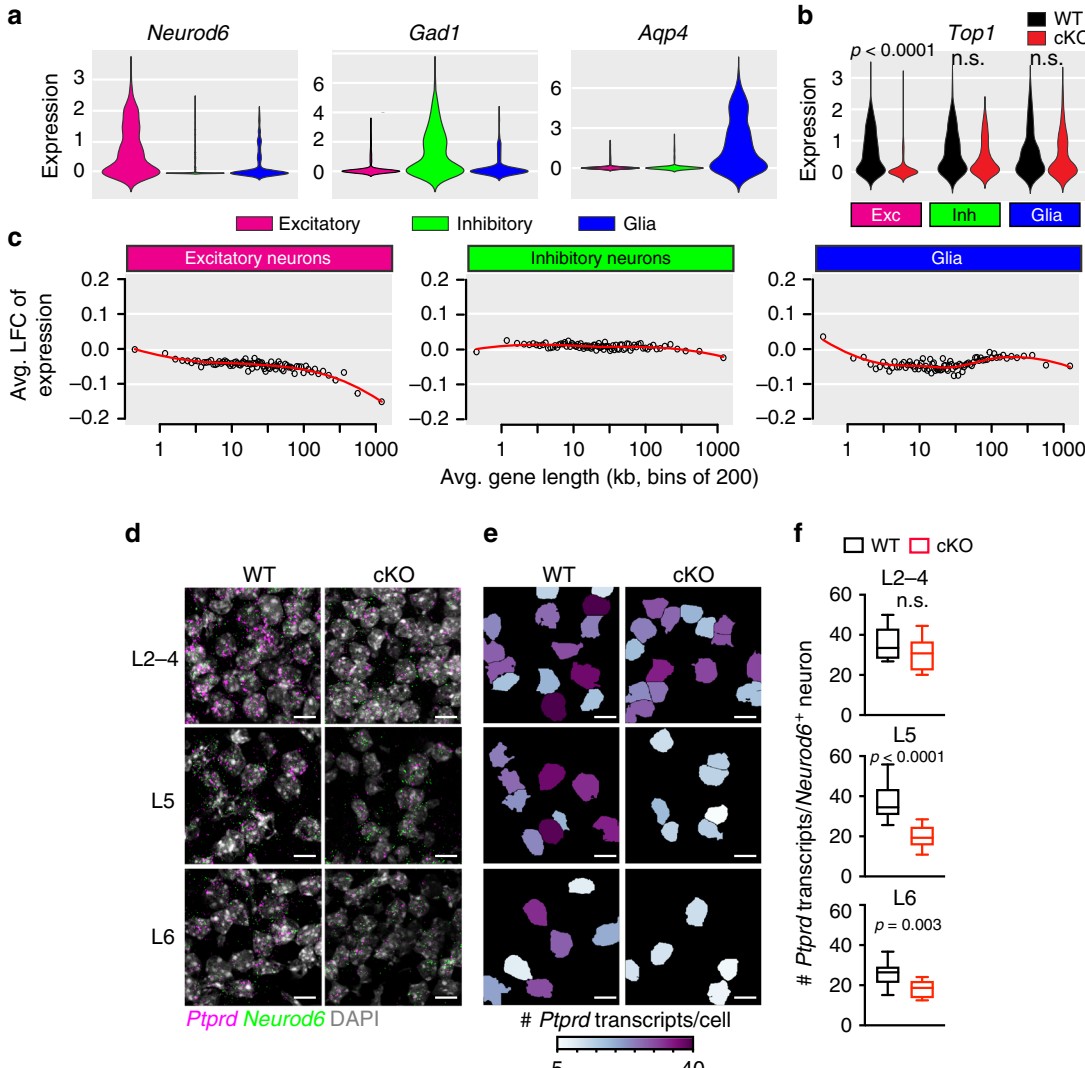

**Fig. 5 Biased downregulation of long genes in lower layer excitatory neurons of *Top1* cKO mice. a** Expression levels of canonical marker genes, *Neurod6*, *Gad1*, and *Aqp4*, in the three main cell clusters identified by single-cell RNA-seq in P7 WT and cKO cortex (*n* = 2 mice/genotype). **b** *Top1* expression levels in WT and cKO cells in each cell cluster. DESeq2 adj. **c** Mean expression change in bins of 200 genes by length in the different cell clusters. **d** In situ hybridization of *Ptprd* and *Neurod6* transcripts in different cortical layers of P7 WT and cKO mice. Sections were also stained with DAPI. Scale bar = 10 μm. Images are representative of two mice per genotype and a total of 11 WT sections and 14 cKO sections. **e** CellProfiler output from analyses of the images shown in **d**. *Neurod6*⁺ neurons were colored based on the number of *Ptprd* transcripts identified. **f** Number of *Ptprd* transcripts/*Neurod6*⁺ neurons/section in cortical neuron layers of WT and cKO P7 cortex. Box is first to third quartile, line is median, error bars are minimum and maximum values, ⁿ·ˢ·*p* > 0.05. Two-sided Student's *t* test (*n* = 11 WT sections; *n* = 14 cKO sections; *n* = 2 mice per genotype). n.s. not significant.

was detected in cKO neurons (24/123 in WT; 60/169 in cKO) (Fig. 6d, Supplementary Data 3), further supporting the previous results. These data suggest that TOP1 plays a key role in maintaining genomic integrity in postmitotic neurons.

**NAD⁺ supplementation reduces neuronal death in *Top1* cKO mice.** PARP1 catalyzes the addition of poly-ADP ribose (PAR) to multiple substrates and consumes NAD⁺ in the process[18–20]. In DNA damage-induced neurodegenerative diseases, PARP1 is hyperactive and leads to increased PAR levels, energetic breakdown, and cell death[17–20]. Given the extensive DNA damage observed in neurons from *Top1* cKO mice, we next evaluated the extent to which PARP1 levels and activity were changed via western blot analysis for PARP1 and its post-translational modification PAR, in WT and *Top1* cKO P7 cortical lysates. We found an increase in both PARP1 and PAR levels in *Top1* cKO cortical lysates relative to WT lysates (Fig. 7a, b, Supplementary Fig. 11),

suggesting that deletion of *Top1* causes an increase in PARP1 activity.

NAD⁺ supplementation was found to increase neuronal survival in models of neurodegeneration and to restore NAD⁺ levels in response to PARP1 hyperactivation[18,43]. Therefore, we hypothesized that increasing NAD⁺ levels may limit neurodegeneration in *Top1* cKO mice. We first evaluated the extent to which two precursors of NAD⁺ synthesis, nicotinamide mononucleotide (NMN) and nicotinamide riboside (NR), elevated NAD⁺ in the neonatal mouse brain. We found that only NR elevated NAD⁺ levels in the brain of P3 mice when injected intraperitoneally (i.p.) (Supplementary Fig. 7). To measure the amounts of NAD⁺ in WT and *Top1* cKO mice and test whether daily injections of NR could maintain high levels of NAD⁺ over time, we injected i.p. WT and *Top1* cKO mice with saline (control) or NR (200 mg/kg per day) beginning on P1 and quantified NAD⁺ levels in cortical extracts at P7 and P15

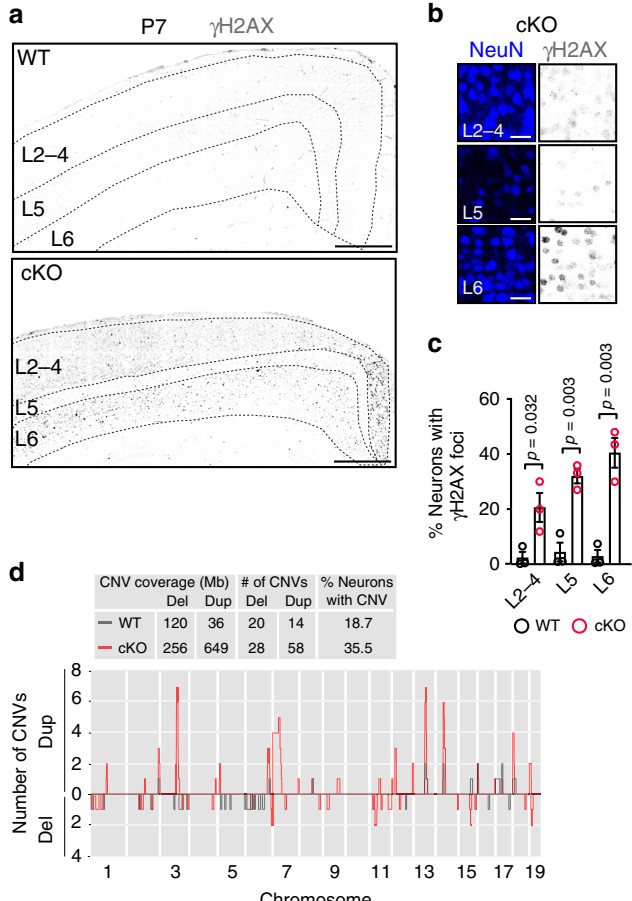

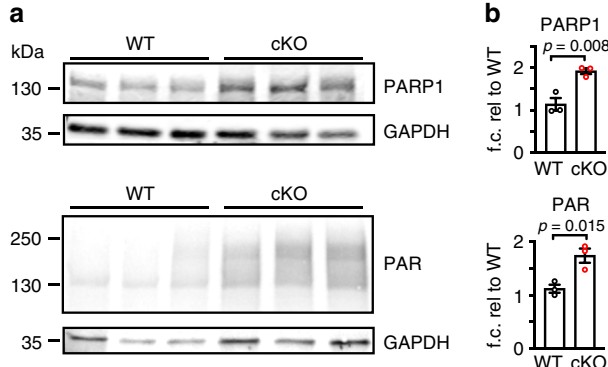

**Fig. 6 Elevated DNA damage and more abundant CNVs in *Top1* cKO mice.**
**a** P7 somatosensory cortex immunostained for γH2AX. Dashed lines delineate the cortical layering identified by NEUN (not shown). Scale bar = 300 μm, **b** insets of P7 *Top1* cKO cortical layers immunostained with NEUN and γH2AX, showing γH2AX foci. Scale bar = 20 μm. Images are representative of three independent experiments for a total of three mice per genotype. **c** Percentages of neurons with γH2AX foci in different layers of WT and cKO P7 cortex. Values are expressed as mean and error bars are ±SEM. Two-sided Student's *t* test. *n* = 3 mice per genotype. **d** Graph showing the overlap of all CNVs identified in WT and cKO neurons at the different genomic locations. *Y*-axis indicates the number of CNVs identified at the correspondent genomic location; above zero are duplications (dup), below zero are deletions (del). (Inset) CNV coverage, number of CNVs identified, and percentages of neurons with CNVs in WT and cKO mice.

**Fig. 7 Increased PARP1 and PAR levels in *Top1* cKO cortex. a** Western blot analysis and **b** quantification of PARP1 and PAR levels in P7 cortical lysates from three mice per genotype. GAPDH was used as a loading control. Protein levels are normalized to GAPDH and expressed as mean fold change relative to WT. Error bars are ±SEM. Two-sided Student's *t* test. *n* = 3 mice per genotype, three technical replicates.

(Table 1). At both time points, NR-treated WT and *Top1* cKO mice showed an increase in NAD$^+$ levels compared to saline-treated mice (Table 1), confirming that NR was efficiently delivered to the postnatal brain with chronic treatment.

NAD$^+$ cortical levels in *Top1* cKO mice were comparable to WT controls at P7 (94.8 ± 2.4 in P7 WT saline vs. 94.7 ± 6.2 in P7 *Top1* cKO saline) (Table 1), consistent with the fact that only a small number of cortical neurons were undergoing energetic breakdown and apoptosis at this age (Fig. 4a, b, Supplementary Fig. 3a, b). Untreated WT mice showed an ~44% increase in NAD$^+$ levels in the cortex from P7 to P15 (94.8 ± 2.4 in P7 WT saline vs. 136.7 ± 10.9 in P15 WT saline). In contrast, NAD$^+$ levels did not increase from P7 to P15 in *Top1* cKO cortical samples (94.7 ± 6.2 in P7 *Top1* cKO saline vs. 92.8 ± 6.6 in P15 *Top1* cKO saline) and showed an ~32% decrease relative to WT controls at P15 (136.7 ± 10.9 in P15 WT saline vs. 92.8 ± 6.6 in

P15 *Top1* cKO saline). Treatment with NR for 15 days restored NAD$^+$ levels (136.7 ± 10.9 in P15 WT saline vs. 155.5 ± 22.9 in P15 *Top1* cKO NR) (Table 1). These data indicated that *Top1* cKO mice developed a deficit in cortical levels of NAD$^+$ by P15 and suggested that cortical NAD$^+$ levels could be restored by supplementation with NR.

We next evaluated the extent to which long-term NAD$^+$ supplementation rescued motor defects and delayed premature death of *Top1* cKO mice. NR treatment did not rescue the motor deficits of *Top1* cKO mice (Fig. 8a), but NR treatment did increase the median survival of *Top1* cKO mice by 5 days (18 days in saline injected vs. 23 days in NR treated), which corresponds to a 30% increase in lifespan (Fig. 8b).

To determine if NR treatment also reduced neuroinflammation and cell death in *Top1* cKO mice, we immunostained NR-treated *Top1* cKO P7 brain sections for cl-CASP3, TUNEL, GFAP, and IBA1. We found that cl-CASP3 activation and TUNEL staining were greatly reduced throughout the somatosensory cortex and hippocampus in NR-treated mice (Fig. 8c). Neuroinflammation was also strongly reduced, as evidenced by reduced IBA1 and GFAP staining (Fig. 8c). NR treatment also modestly increased hippocampal area, increased overall cortical thickness, and increased lower layer thickness in cKO mice at P15 (Fig. 8d, e). Moreover, we observed a 2- to 3-fold increase in the number of CTIP2$^+$ neurons in lower cortical layers and in the hippocampus of *Top1* cKO mice treated with NR (Fig. 8d, e). These data suggest that neuron loss is only partially dependent on NAD$^+$ depletion.

**p53 loss partially rescues neurodegeneration in *Top1* cKO mice.** We asked whether partial or complete deletion of *p53* could rescue neuron death in *Top1* cKO mice. To answer this question, we crossed *Neurod6*$^{Cre/+}$::*Top1*$^{fl/fl}$ mice with *p53* KO mice[44] to generate WT and *Top1* cKO mice with *p53* WT, HET, or KO alleles. Mice were assayed for motor tests at P7 and P12, brains were collected at P15 and cortical thickness was quantified by staining with CUX1 and CTIP2. *p53* deletion failed to rescue motor deficits (Fig. 9a) and did not extend lifespan in a small cohort (*n* = 2) of *Top1* cKO::*p53* KO animals, which died in the third postnatal week. However, both homozygous and HET loss of *p53* led to an ~1.5-fold increase in cortical thickness and an ~2.5-fold increase in hippocampal area in *Top1* cKO mice (Fig. 9b, c), bringing these areas to ~60% of *Top1* WT size. The rescue observed in *Top1* cKO::*p53* KO and *Top1* cKO::*p53* HET cortex was caused by an ~3-fold increase in L5 thickness and an

**Table 1 Quantification of NAD$^+$ cortical levels in WT and cKO mice treated with saline or NR.**

| | Mean | SEM | P7 WT sal | P7 cKO sal | P7 WT NR | P7 cKO NR | P15 WT sal | P15 cKO sal | P15 WT NR |
|---|---|---|---|---|---|---|---|---|---|
| P7 WT sal | 94.8 | 2.4 | — | | | | | | |
| P7 cKO sal | 94.7 | 6.2 | 0.336[a] | — | | | | | |
| P7 WT NR | 159.4 | 7.2 | **** | **** | — | | | | |
| P7 cKO NR | 146.3 | 3.8 | *** | *** | 0.108[a] | — | | | |
| P15 WT sal | 136.7 | 10.9 | 0.007 | 0.007 | 0.086[a] | 0.264[a] | — | | |
| P15 cKO sal | 92.8 | 6.6 | 0.314[a] | 0.314[a] | *** | *** | 0.008 | — | |
| P15 WT NR | 240.7 | 25.4 | **** | **** | **** | **** | **** | **** | — |
| P15 cKO NR | 155.5 | 22.9 | *** | *** | 0.308[a] | 0.184[a] | 0.137[a] | *** | **** |

Mean plus/minus SEM of NAD$^+$ levels (pmol/mg) in cortical lysates of P7 and P15 WT and cKO mice treated with saline (sal) or NR beginning at P1. Adjusted $p$ values for each comparison are shown ([a]= non-significant $p > 0.05$, ***adj. $p < 0.001$, ****adj. $p < 0.0001$). Three-way ANOVA with false discovery rate correction (two-stage step-up method of Benjamini, Krieger, and Yekutieli, $Q = 0.05$). For P7: $n = 6$ WT;saline mice, $n = 5$ WT;NR mice, $n = 6$ cKO;saline mice, $n = 7$ cKO;NR mice. For P15: $n = 3$ WT;saline mice, $n = 4$ WT;NR mice, $n = 4$ cKO;saline mice, $n = 3$ cKO;NR mice.

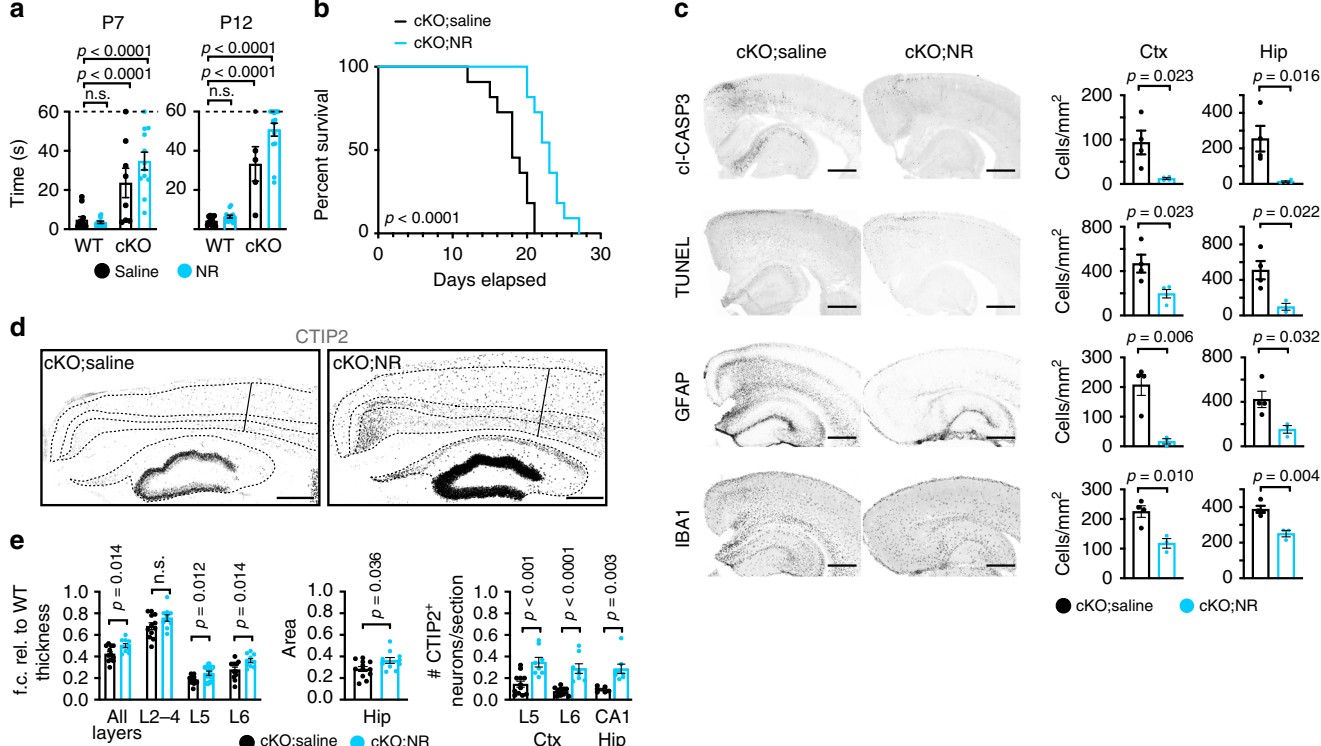

**Fig. 8 NAD$^+$ supplementation with nicotinamide riboside partially blocks neurodegeneration and increases lifespan of *Top1* cKO mice. a** P7 righting assay (left) and P12 geotaxis (right) performed on WT and *Top1* cKO mice treated with saline or NR 200 mg/kg per day from P1 onward. For P7: $n = 12$ WT;saline mice, $n = 12$ WT;NR mice, $n = 8$ cKO;saline mice, $n = 12$ cKO;NR mice. For P12: $n = 14$ WT;saline mice, $n = 14$ WT;NR mice, $n = 5$ cKO;saline mice, $n = 14$ cKO;NR mice. Dashed lines represent cutoff time. n.s.$p > 0.05$. One-way ANOVA with Dunnett's multiple comparison test. **b** Kaplan–Meier survival curve of *Top1* cKO mice treated with saline or 200 mg/kg per day NR from P1 onwards. Mantel–Cox test. WT = 11, cKO = 10 mice. **c** *Top1* cKO mice were treated with saline or 200 mg/kg per day NR for 6 days starting from P1 and then brains were collected. Sections containing cortex (Ctx) and hippocampus (Hip) were immunostained and for cl-CASP3, TUNEL, GFAP, and IBA1 (left panel), scale bar = 500 μm. The number of cells positive for each marker was quantified (right panel). n.s.$p > 0.05$. Two-sided Student's $t$ test. For cl-CASP3 quantification $n = 4$ mice per condition, for TUNEL, GFAP, and IBA1 quantifications $n = 4$ cKO;saline mice, $n = 3$ cKO;NR mice. **d** CTIP2 staining of cerebral cortex and hippocampus of P15 *Top1* cKO mice treated with saline or 200 mg/kg per day NR for 14 days. Scale bar = 300 μm. Dashed lines delineate cortical layering and hippocampus based on NEUN, CUX1, and CTIP2 staining. The solid line indicates the area used to quantify cortical thickness. Images are representative of at least three independent experiments for a total of three mice per genotype. **e** Hippocampal area and number of CTIP2$^+$ cells per section of cortex and hippocampus. n.s.$p > 0.05$. Two-sided Student's $t$ test. For thickness and area quantifications: $n = 3$ WT mice, $n = 6$ cKO;saline mice, $n = 5$ cKO;NR mice. For L5-6 CTIP2$^+$ neurons quantification: $n = 3$ WT mice, $n = 6$ cKO;saline mice, $n = 4$ cKO;NR mice. For CA1 CTIP2$^+$ neurons quantification: $n = 3$ WT mice, $n = 3$ cKO;saline mice, $n = 4$ cKO;NR mice. Two sections per mouse were used for all quantifications. Values are mean and error bars are ±SEM. n.s. not significant.

~2-fold increase in L6 thickness, bringing L5 from ~18% to ~52% of *Top1* WT L5 size and L6 from ~28% to 55% of *Top1* WT L6 size (Fig. 9b, c). Both homozygous and HET loss of *p53* in *Top1* cKO mice led to an increase in the number of CTIP2$^+$ neurons in lower cortical layers and the CA1 hippocampal region.

We observed a complete rescue in the number of CTIP2$^+$ neurons of the CA1 hippocampal region in *Top1* cKO::*p53* KO mice (Fig. 9b, c). To test whether the increase in cortical and hippocampal size in *Top1* cKO::*p53* KO mice was due to an increase in the number of TOP1$^-$ neurons, we performed TOP1 and NEUN

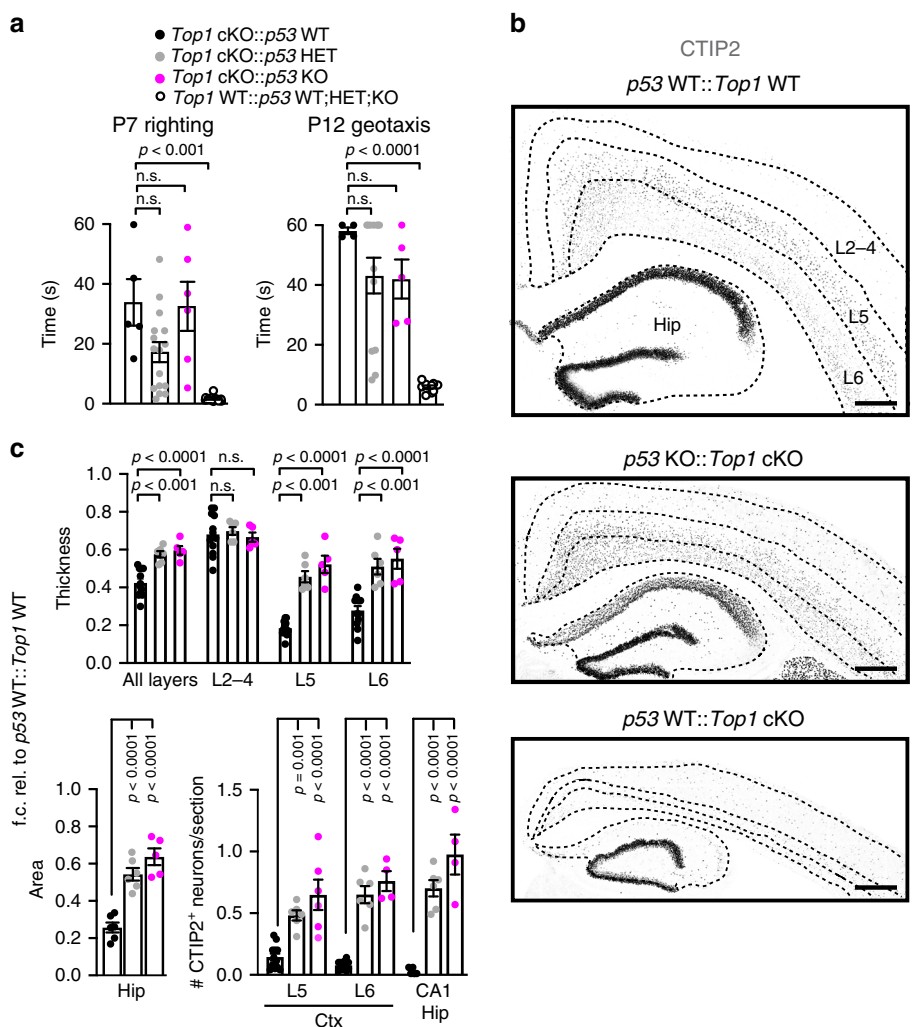

**Fig. 9 Deletion of *p53* partially blocks neurodegeneration but does not rescue behavioral deficits in *Top1* cKO mice. a** Righting assay performed at P7 and geotaxis assay performed at P12 with the indicated genotypes. n.s.$p > 0.05$. One-way ANOVA with Dunnett's multiple comparison test. For P7 $n = 5$ *p53* WT::*Top1* cKO mice, $n = 16$ *p53* HET::*Top1* cKO mice, $n = 6$ *p53* KO::*Top1* cKO mice, $n = 10$ *p53* WT;HET;KO::*Top1* WT mice. For P12 $n = 4$ *p53* WT::*Top1* cKO mice, $n = 13$ *p53* HET::*Top1* cKO mice, $n = 5$ *p53* KO::*Top1* cKO mice, $n = 10$ *p53* WT;HET;KO::*Top1* WT mice. **b** Cerebral cortex and hippocampus from P15 mice. Genotypes of the *Top1* and *p53* alleles are indicated. Sections were immunostained for CUX1, CTIP2, and NEUN. Dashed lines are based on NEUN, CUX1, and CTIP2 staining and delineate cortical layering and hippocampus. Only CTIP2 staining is shown. Scale bar = 300 μm. Images are representative of at least three independent experiments for a total of at least three mice per genotype. **c** Quantifications of cortical thickness, hippocampal area, and number of CTIP2+ neurons per section in cortical layers and the CA1 region of the hippocampus. n.s.$p > 0.05$. One-way ANOVA with Dunnett's multiple comparison test for each cortical layer of hippocampal area. For cortical thickness quantifications: $n = 6$ *p53* WT::*Top1* cKO mice, $n = 3$ *p53* HET::*Top1* cKO mice, $n = 3$ *p53* KO::*Top1* cKO mice. Two sections per mouse. For hippocampal area quantifications: $n = 3$ *p53* WT::*Top1* cKO mice, $n = 3$ *p53* HET::*Top1* cKO mice, $n = 3$ *p53* KO::*Top1* cKO mice. Two sections per mouse. For L5 CTIP2 quantifications: $n = 6$ *p53* WT::*Top1* cKO mice, $n = 3$ *p53* HET::*Top1* cKO mice, $n = 3$ *p53* KO::*Top1* cKO mice. Two sections per mouse. For L6 CTIP2 quantifications: $n = 6$ *p53* WT::*Top1* cKO mice, $n = 3$ *p53* HET::*Top1* cKO mice, $n = 2$ *p53* KO::*Top1* cKO mice. Two sections per mouse. For CA1 CTIP2 quantifications: $n = 3$ *p53* WT::*Top1* cKO mice, $n = 3$ *p53* HET::*Top1* cKO mice, $n = 2$ *p53* KO::*Top1* cKO mice. Two sections per mouse. Values are mean and error bars are ±SEM. n.s. not significant.

immunostaining of the cortex and hippocampus. While very few TOP1−/NEUN+ neurons were observed in the lower cortical layers and CA1 hippocampal region of *Top1* cKO::*p53* WT mice, these areas were almost completely composed of TOP1−/NEUN+ neurons in *Top1* cKO::*p53* KO (Supplementary Fig. 8). These data strongly suggest that neurodegeneration caused by *Top1* loss is partially p53 dependent.

## Discussion

Our data support a model whereby loss of TOP1 causes transcriptional stress in postmitotic neurons that leads to high levels of DNA damage, subsequent activation of DNA damage-induced

cell death pathways and neurodegeneration (Supplementary Fig. 9). Interestingly, aberrant TOP1 activity has been shown to cause a similar phenotype in neurodegenerative genome instability syndromes[45,46]. In these pathologies, the formation of stable TOP1cc's at active transcription sites induces transcriptional arrest, accumulation of RNA:DNA hybrids (R-loops), and the formation of dsDNA breaks in postmitotic cells[46,47]. TOP1 deficiency has also been linked to transcriptional stress, accumulation of R-loops, and genomic instability in mitotic cells[38,40,48,49]. DNA damage caused by TOP1 depletion was attributed to the collision between transcriptional machinery and the DNA replication fork[40]. Our data suggest that TOP1 loss induces DNA damage accumulation independent of DNA

replication in postmitotic neurons, possibly because of increased DNA torsional stress and R-loop formation during transcription. Further investigation will be required to understand the extent to which R-loops accumulate in *Top1* cKO neurons. Altogether, these data indicate that TOP1 maintains genomic integrity in the central nervous system.

Another interesting relationship between *Top1* deletion and TOP1cc formation is the downstream activation of multiple DNA damage-induced pathways, including p53 activation[45] and increased PARP1 activity[17]. Consistent with other forms of neurodegeneration that feature DNA damage, *Top1* cKO mice show increased PARP1 activity, reduced NAD$^+$ levels, neuroinflammation, and neuronal death[18–20]. Our study suggests that *Top1* cKO mice could be used to further understand the pathways involved in DNA damage-induced neurodegeneration in vivo, but on a greatly accelerated time scale.

We also found that treatment with NR restored NAD$^+$ levels in the brain of *Top1* cKO mice. This treatment increased lifespan by 30%, significantly reduced neuroinflammation, and reduced neuron death, including CTIP2$^+$ neurons in the lower cortical layers and hippocampus. These results corroborate other studies indicating that NAD$^+$ supplementation can extend lifespan, decrease neuroinflammation, and delay neuron death in neurodegeneration models, including in AD and DNA-repair-deficient mouse models[18,50–52]. However, given that the cortical and hippocampal reduction were only modestly rescued by NR treatment, NAD$^+$-independent pathways must also contribute to neurodegeneration in *Top1* cKO mice.

In support of this possibility, genetic ablation of *p53* led to a greater rescue in cortical thickness and hippocampal area compared to NR treatment and led to a near complete rescue in the number of CTIP2$^+$ projecting neurons. The tumor suppressor p53 is the main driver of DNA damage-induced death in neurons and is a common downstream effector of multiple cellular stress sensors, including PARP1[20,53–55]. However, neither p53 depletion nor NAD$^+$ supplementation rescued the motor phenotypes and premature death observed in *Top1* cKO mice. These data suggest that loss of TOP1 could have additional effects that compromise neuron function, such as elevating somatic mutation burden (Fig. 6d), transcriptional downregulation of long synaptic genes[23,24,26] (Fig. 5c–f), which attenuates synaptic activity[24], impaired RNA splicing[56], and/or impaired ribosomal RNA biosynthesis[49,57]. Thus, transcriptional decline and somatic mutations could incapacitate neurons well before cell loss is apparent.

Selective vulnerability of different neuronal populations to cell death has been described in many neurodegenerative diseases and it has often been attributed to intrinsic properties of the affected neurons[58]. Our data indicate that among cortical neurons, lower layer cortical neurons are particularly sensitive to *Top1* loss. Using single-cell RNA-seq and single-molecule in situ hybridization, we found that *Top1* loss strongly impaired expression of long genes, with a bias for lower layer neurons. Many long genes encode proteins that play key roles in synapse formation and function. Downregulation of these genes is likely to compromise neuron physiology, as we previously found[26], and health independent of DNA damage.

Alternatively, lower layer neurons could be more vulnerable because of their earlier birthdate and presumably earlier expression of *Neurod6$^{Cre}$*. Cortical layers are produced in an inside-out order, where lower layers are produced earlier than upper layers[29]. Therefore, lower layer neurons have more time to accumulate transcriptional stress, DNA damage, and somatic mutations that affect neuronal functions and ultimately cause neurodegeneration. In support of this hypothesis, we observed that lower layer cortical neurons present higher levels of DNA

damage and downregulation of long genes in *Top1* cKO mice. However, countering this possibility, we found that L5 neurons are more vulnerable to *Top1* deletion than L6 neurons, yet L5 neurons are born after L6 neurons.

The unique properties of L5 motor neurons, such as large soma size and the ability to establish extremely long distance axonal projections (requiring adhesion molecules that are encoded by long genes), are known to be more vulnerable to cell death[36,59]. Projection neurons that maintain a large soma, lengthy projections, and higher transcriptional output of long genes, likely consume more energy than local projection neurons. L5 upper motor neurons are thus likely to be more sensitive to NAD$^+$ depletion and energetic breakdown than other neuronal populations. Interestingly, ALS and other motor neuron neurodegenerative diseases have been associated to defects in R-loop resolution[60], providing another possible explanation for why this neuronal population is more sensitive to *Top1* loss. Ultimately, future studies will be needed to characterize the effect of *Top1* deletion in other neuronal subtypes, including inhibitory neurons and neurons from other brain regions.

## Methods

**Mice**. All animal procedures were approved by the University of North Carolina at Chapel Hill Animal Care and Use Committee. Mice were housed at temperatures of 18–23 °C with 40–60% humidity with food and water provided ad libitum and maintained on a 12-h dark/light cycle. *Top1$^{fl/fl}$* conditional mice were previously described[24]. *Neurod6$^{Cre}$* mice[28] and *p53$^{−/−}$* mice[44] were obtained from Jackson Laboratory. Genomic DNA extracted from tail or ear samples was utilized for mouse genotyping by PCR using standard techniques. Primers for gene amplification are as follows (listed 5′–3′): *Top1* geno2 GAGTTTCAGGACAGCCAGGA and *Top1* geno3 GGACCGGGAAAAGTCTAAGC amplifying *Top1* WT and KO and floxed alleles, CreF GATGGACATGTTCAGGGATCGCC and CreR CTCCCATCAGTACGTGAGAT amplifying the *Cre* allele; p53 WT-F CCCGAGTATCTGGAAGACAG and p53 WT-R ATAGGTCGGCGGTTCAT amplifying *p53* WT allele; p53 ex7 TATACTCAGAGCCGGCCT and p53 Neo TCCTCGTCGTTTACGGTATC amplifying *p53* KO allele.

**Motor assays**. For the righting assay, P7 pups were placed on their backs on a sheet of paper and held in position for 3 s. Pups were then released and the total time needed to flip on their feet was recorded. For the negative geotaxis assay, P12 pups were held for 5 s on a 45° incline with the head pointing downward. Pups were then released and the total amount of time needed to face upward was recorded. For both assays, a cutoff maximum of 60 s was given for every trial. Pups were tested three times with a break of 30 s between each trial. The times recorded for the three trials were then averaged. *Top1* cKO and cHET mice were compared with WT littermates or age-matched mice. Males and females mice were used for motor assessments. GraphPad Prism (version 7.00) was used for statistical analysis.

**Tissue collection and immunostaining**. At the indicated time points, mice were anesthetized and perfused first with phosphate-buffered saline (PBS) and then with 4% paraformaldehyde (PFA). Brains were dissected, post fixed in 4% PFA overnight at 4 °C, incubated in 30% sucrose at 4 °C, and embedded in M-1 embedding matrix (Thermo Scientific, 1310). Brains were cut on a cryostat and free-floating sections were collected and stored in a freezing solution consisting of PBS, ethylene glycol, and glycerol. Sections were rinsed in PBS, treated with citrate buffer for 30 min at 95 °C, rinsed again in PBS followed by rinses with Tris-buffered saline containing Triton-X-100 (TBS/TX, 0.05 M Tris, 2.7% sodium chloride, 0.3% Triton-X-100, pH 7.6). Sections were blocked in 10% normal donkey serum in TBS/TX (NDS/TBS/TX) for 1 h and then incubated overnight at room temperature in primary antibody cocktails diluted in NDS/TBS/TX. Primary antibodies used were rabbit anti-TOP1 (1:300; GeneTex, GTX63013, EPR5375), guinea-pig anti-NEUN (1:400; EMD-Millipore, ABN90P), rabbit anti-CUX1 (1:200; Santa Cruz Biotechnology, sc-13024), rat anti-CTIP2 (1:200; Abcam, ab18465), rabbit anti-cleaved-CASP3 (1:100; Cell Signaling Technology, 9664), rabbit anti-IBA1 (1:400; Wako, 019-19741), goat anti-GFAP (1:750; Abcam, ab53554), rabbit anti-phospho-Histone H2A.X (1:50; Cell Signaling Technology, 2577), and phospho-53BP1 (Ser1778) (1:50, Cell Signaling Technology, 2675).

After incubation with primary antibodies, sections were rinsed in TBS/TX and blocked with NDS/TBS/TX for 30 min. All secondary antibodies were diluted at 1:200 in NDS/TBS/TX and applied for 6 h. These included donkey anti-rabbit IgG Alexa 488 or Alexa 568 (Thermo Fisher Scientific, A21206 and A10042, respectively), donkey anti-goat IgG-Cy3 (Jackson ImmunoResearch Laboratories, 705-165-003), donkey anti-rat IgG-Cy3 (Jackson ImmunoResearch Laboratories, 712-165-153), and donkey anti-guinea-pig IgG Alexa 647 (Jackson ImmunoResearch Laboratories, 706-605-148). DAPI (1:4000; Fisher Scientific,

EN62248) was added to secondary antibody cocktails. Sections were then rinsed in TBS/TX and PBS. Detection of apoptotic cells by TUNEL assay was performed on previously immunostained tissue sections using the DeadEnd™ Fluorometric TUNEL System (Promega, G3250) following the manufacturer's instructions. All stained sections were mounted onto SuperFrost Plus slides (Fisher Scientific, 12-550-15) and coverslipped with FluoroGel mounting medium (Electron Microscopy Sciences, 17985-10). Imaging was performed on a Zeiss LSM 710 confocal laser scanning microscope.

**Image analysis and quantitation.** Confocal images of somatosensory cortex were collected from individual brain sections for each animal. At least two animals per condition and two sections per animal were used for every quantification. For neocortical layer-thickness analysis, images were transferred to ImageJ and individual layer boundaries were determined by the changes in density and appearance of NEUN, CUX1, and CTIP2 labeling. Layer thickness was measured using the "straight line" tool present in ImageJ. Measurements were expressed as absolute numbers or normalized to control samples and expressed as fold changes. For cell density and cell area quantification of CTIP2+, cl-CASP3+, TUNEL+, IBA1+, and GFAP+ cells a general pipeline was written on ImageJ and parameters were adapted based on the intensity and distribution of the specific cell-type markers. Briefly, brain sections where co-stained with the cell marker of interest and NEUN. Images of the cortex were manually cut into the different layers based on NEUN labeling. CUX1 and CTIP2 were also used to distinguish different layers. The channel of the cell marker of interest was then selected and a pixel-intensity threshold was set manually by an observer blind to the genotype. The same pixel-intensity threshold for every marker was applied to all layers and all genotypes. For CTIP2[high] cell density analysis, a threshold that allows the visualization of only highly expressing CTIP2 cells of L5 was chosen. After watershed segmentation, the binary image mask was analyzed using the particle analysis tool to count the number of events. The total number of particles and particle area were then measured using the ROI Manager tool in ImageJ. The total number of cells identified within a cortical layer was divided by the area of that layer.

To measure the percentage of neurons with γH2AX foci, we immunostained brain sections for NEUN and γH2AX. Images were processed in ImageJ and a layer-specific mask on NEUN+ cells was created as described above. Then, the γH2AX signal was selected and the number of γH2AX foci was determined using the "find maxima" function in ImageJ. Briefly, a noise-tolerance threshold that allows the detection γH2AX foci was selected by an observer blind to the genotype and applied to all samples. A binary image of γH2AX foci was created through the single points option. The NEUN+ cell mask was then overlapped to the γH2AX foci binary image using the ROI Manager tool and the RawIntDen within each neuron was calculated. The RawIntDen was then divided to the value of a single pixel (255) in order to calculate the number of γH2AX foci per neuron. Representative images have been cropped and equally adjusted for brightness and contrast in ImageJ for presentation. GraphPad Prism (version 7.00) was used for statistical analysis.

**PET imaging with [18]F-PBR111.** PET imaging probe of PBR ligand, [18]F-PBR111, was produced at the UNC BRIC Cyclotron and Radiochemistry Core facility according to a well-established radiosynthesis method[61]. Radiochemical purity was >95% confirmed from the HPLC assay. Animal PET/CT imaging was conducted in the UNC Small Animal Imaging facility using a small animal PET/CT scanner (Argus, SEDECAL Inc., Spain) with a spatial resolution of 1.2 mm in the center field of view. Animals were anesthetized with isoflurane inhalation (1.5–2.5% isoflurane mixed with oxygen). A catheter was placed in the abdominal region for i. p. injection of radiotracer. WT and *Top1* cKO P15 mice were placed on the imaging bed. [18]F-PBR111 (~5 MBq in 50 μl) was administered to each animal through an i.p. catheter, followed by a 60-min dynamic scan. The computed tomography (CT) scan was conducted before the PET scan for attenuation correction and structure reference. Dynamic data were binned into six frames with 10 min/frame. Images were reconstructed using the 2D-OSEM (two-dimensional-ordered subset expectation maximization) algorithms with scatter, attenuation, and decay correction. Standardized uptake value was calculated by normalizing the signal to the injection dose and animal body weight. After PET/CT images, brains were collected and autoradiography was conducted using a digital phosphor imaging system (Cyclone Plus storage phosphor system, PerkinElmer Inc.). GraphPad Prism (version 7.00) was used for statistical analysis.

**Drop-seq procedure.** P7 mice were decapitated and cortices were isolated, cut into pieces with a sterile razor blade, and incubated in papain diluted in standard artificial cerebrospinal fluid (aCSF) media (NaCl 124 mM, KCl 2.5 mM, NaH$_2$PO$_4$ 1.2 mM, NaHCO$_3$ 24 mM, HEPES 5 mM, glucose 13 mM, MgSO$_4$-7H$_2$O, 2 mM, CaCl$_2$-2H$_2$O, 2 mM, pH 7.3) supplemented with (2*R*)-amino-5-phosphonovaleric acid (APV) and tetrodotoxin (TTX) at 37 °C for 30 min. Cortical pieces were then washed and gently dissociated in aCSF media with APV and TTX. Cortical cells were then filtered, captured, and processed into single-cell transcriptomes using Drop-seq[62]. Complementary DNA libraries were sequenced on two lanes of a HiSeq2500 on Rapid Run mode. FASTQ files were processed using the Drop-seq Toolkit 1.2 where possible and aligned to the mouse genome (mm10) using

STAR[63]. Barcoded bead synthesis errors were corrected using a publicly available command line script[64].

**Single-cell RNA-seq bioinformatics.** Cells expressing fewer than 300 different genes or whose mitochondrial transcripts exceeded 10% were excluded. Genes expressed in fewer than three cells were removed. The resulting cell × gene matrix was median centered and log normalized to generate the final gene expression matrix (1596 cells × 11,112 genes). Louvain clustering was performed on the top 48 principal components (based on eigenvalues greater than randomly permutated data [$n = 1000$]) using a silhouette score-optimized nearest-neighbor parameter of (67 nearest neighbors[27]). Cluster-specific marker genes were detected using a presence–absence binomial test (mean log$_2$ fold change >1.0, $p$ value < 0.05). Differentially expressed genes were identified using DESeq2 (adj. $p$ value < 0.05[65]). To generate plots showing change in expression by gene length, genes were ordered by gene length and binned into groups of 200. The average fold change (calculated by DESeq2) was plotted against the average gene length for each bin in each cluster.

**Single-molecule in situ hybridization and image analysis.** RNAscope fluorescent multiplex assay was performed following the manufacturer's instructions with the following changes. To avoid tissue deterioration, unfixed frozen sections of P7 cortex were fixed in fresh 4% PFA for 20 min and protease treatment was reduced to 10 min. The following probes were obtained from ACDbio: Mm-*Neurod6*-C3 (cat. no. 444851-C3), Mm-*Ptprd* (cat. no. 474651). Images from two mice per genotypes ($n = 11$ WT sections, $n = 14$ cKO sections) were acquired on a Zeiss LSM 710 confocal laser scanning microscope at ×40. Multichannel images were opened in ImageJ and manually separated into different layers based on DAPI labeling and separated in the different channels. *Ptprd* and *Neurod6* channels were then selected and a pixel-intensity threshold was set manually to normalize the background among images of the different genotypes. A pixel-intensity threshold was also applied to DAPI images and followed by watershed segmentation. The observer was blind to the genotype. Binary images were then processed on CellProfiler (https://cellprofiler.org/) using a custom-designed pipeline. Briefly, single cells and *Neurod6* transcripts were identified as objects using the corresponding channels. Then, *Neurod6* transcripts were assigned to each cell as children objects and quantified. *Neurod6*+ cells were selected by applying a threshold of 5 *Neurod6* transcripts per cell. Single *Ptprd* transcripts or clusters were distinguished using a size threshold and were each assigned to *Neurod6*+ cells. Images showing the number of *Ptprd* transcripts/*Neurod6*+ cells was then generated by the DisplayDataOnImage function. The average number of single *Ptprd* transcripts or *Ptprd* clusters/*Neurod6*+ cell/section was the calculated. The final number of *Ptprd* transcripts/*Neurod6*+ cell/section was calculated with the following formula: # *Ptprd* transcripts/*Neurod6*+ cell/section = number of single *Ptprd* transcripts/*Neurod6*+ cell + (#*Ptprd* clusters/*Neurod6*+/section divided by the average area of single *Ptprd* transcripts/section). GraphPad Prism (version 7.00) was used for statistical analysis.

**CNV analyses of FACS sorted cortical neurons.** P7 cortices were dissected with the hippocampus removed and frozen at −80 °C. Frozen cortices were dissociated mechanically. Nuclei were purified using an iodixanol cushion[66] and stained for NEUN (anti-human NeuN Alexa Fluor 555 conjugate (Millipore, MAB377)) and SYTO 13 green fluorescent nucleic acid stain (Thermo Fisher, S7575). NEUN+/SYTO 13+ neuronal nuclei were isolated by flow sorting into 8-well thin-wall PCR tube strips[42]. For the initial low cell number experiment, WGA was performed using a modified multiple annealing and loop-based (MALBAC) approach[67]. Single-cell libraries of 39 WT neurons and 43 cKO neurons extracted from one mouse per genotype were individually barcoded and prepared for Illumina sequencing using NextFLEX (BIOO Scientific, Austin, TX) following the manufacturer's instructions. Sequencing was performed on an Illumina HiSeq RAPID platform. CNVs were detected using an unbiased CNV detection approach[42]. WGA quality control using Bayesian information criteria (>−1.5) removed 13 cells from subsequent analysis, and copy number states <1.22 and >2.82 were determined in DNAcopy ($α = 0.001$, undo.SD = 0, min. width = 5) segmentation. BWA (version 0.7.12) was used to align Illumina single-end reads to the genome. Samtools (v1.1) was used to generate BAM files. Picard Tools (v1.105) was used to remove duplicate reads. Bedtools (v2.17.0) was used to count aligned reads in genomic bins. R (v3.4.1) was used to normalize binned read counts to copy number estimates, segment bin values (DNAcopy, v1.50.1), and plot CNV profiles (ggplot2, v2.2.1). R was also used to fit Gaussian distributions to dataset-wide segment data to determine CNV thresholds (mixtools, v1.1.0).

For the follow-up experiment, 10× sequencing data from 123 WT and 169 cKO NEUN+/SYTO 13+ sorted cortical neurons derived from one mouse per genotype were aligned to the mouse genome (mm10) using CellRanger-dna cnv (version 1.1.0). Single-cell BED files were extracted from CellRanger output using SAMtools (version 1.9), BAMtools (version 2.5.1), and BEDtools (version 2.15.0), and then were uploaded to Ginkgo[68] for CNV analysis using default parameters. Following Ginkgo CNV analysis, segmentation results were downloaded and filtered to remove likely false-positive CNVs. Regions containing six or more CNVs were considered to be "hotspots" for false-positive CNVs, and any CNVs located entirely within the boundaries of a hotspot were excluded. To further safeguard against

false positives, CNVs under 5 Mb in length were also excluded. Bin level copy number estimates for all single cells were downloaded from Ginkgo and plotted with filtered CNV results in R (version 3.4.1) using ggplot2 (version 2.2.1) to generate CNV profiles.

**Western blotting**. P7 cortices were dissected from both *Top1* cKO and WT control mice and lysed in RIPA buffer (0.05 M Tris-HCl, pH 8, 0.15 M NaCl, 0.5% deoxycholic acid, 1% NP-40, and 0.1% sodium dodecyl sulfate (SDS)), Halt Protease Inhibitor Cocktail (Thermo Fisher, 87785), and Halt Phosphatase Inhibitor Cocktail (Thermo Fisher, 78420). Lysates were sonicated and cleared by centrifugation. Protein concentration was determined using the Pierce BCA Protein Assay Kit (Fisher, 23227). Equal amounts of protein were denatured in reducing sample buffer, separated by SDS-PAGE (polyacrylamide gel electrophoresis) gels, and blotted to nitrocellulose membranes (Bio-Rad). Blots were blocked with Odyssey Blocking Buffer (Fisher, 927-40000) for 1 h at room temperature, and then incubated overnight at 4 °C with primary antibodies. The primary antibodies used were rabbit anti-PARP (1:1000, Cell Signaling, 9532) rabbit anti-PAR/pADPr (1:1000, R&D Systems, 4336-APC-050), and mouse anti-GAPDH (1:1000, Thermo Fisher, MA5-15738). After washing with TBS containing 0.5% Tween 20 (TBS-T), membranes were incubated with goat anti-mouse IRdye 680 (1:10,000 Licor Odyssey, 925-68070) and IRDye 800CW donkey anti-rabbit (1:10,000 Licor Odyssey, 926-32213) diluted in Odyssey Blocking Buffer 1 h at room temperature. Blots were washed with TBS-T and detection was performed with the Odyssey CLx Imaging System (Li-COR) using the Image Studio Software (Odyssey). GraphPad Prism (version 7.00) was used for statistical analysis.

**NR treatment**. Nicotinamide riboside chloride powder (Medkoo, 329479) was diluted to 10 mg/ml stock solution in saline, aliquoted, and stored at −80 °C. *Neurod6^{Cre/+}::Top1^{fl/+}* males were bred with three *Top1^{fl/fl}* females for 48 h. After 14 days, pregnant females were moved in a separate cage and checked for P0 pups twice a day. Twenty-four hours after birth, P1 pups were injected intraperitoneally with 20 μl/g of saline or NR once a day (final 200 mg/kg per day of NR). Entire litters were injected with saline or NR in a blinded manner. At P7 and P12, pups were assayed for righting and geotaxis assays, respectively, and genotyped. Mice were monitored every day. An independent set of litters treated as described was sacrificed at P7 or P15 for immunofluorescence analysis and NAD$^+$ quantification. Brains were collected at the indicated time points of treatment, and after 1 h from the last injection, half of the brain was drop-fixed in 4% PFA for immunostaining and half was divided into pieces and flash-frozen for NAD$^+$ quantification. GraphPad Prism (version 7.00) was used for statistical analysis.

**NAD$^+$ quantifications**. Cortices were dissected from WT and *Top1* cKO mice and flash-frozen in pieces ranging from 15 to 30 mg in weight. Cortical samples were collected 1 h after the last i.p. injection with saline, NR, or NMN. All cortical samples were thawed and processes for NAD$^+$ quantification at the same time using EnzyChrom NAD$^+$/NADH Assay Kit (BioAssay Systems, E2ND-100) following the manufacturer's instructions. NAD$^+$ quantification were collected using Synergy 2 microplate reader (Biotek) and using the Gen5 (version 2.0) software. The resulting NAD$^+$ concentration was multiplied to the final volume of reagents for each sample and divided by the total amount of starting tissue to calculate NAD$^+$ pmol/mg of tissue. GraphPad Prism (version 7.00) was used for statistical analysis.

**Reporting summary**. Further information on research design is available in the Nature Research Reporting Summary linked to this article.

## Data availability

Single-cell genomic data are available from the NCBI Sequence Read Archive (https://www.ncbi.nlm.nih.gov/sra/PRJNA548496). Single-cell RNA-sequencing data are available from the NCBI Gene Expression Omnibus database (https://www.ncbi.nlm.nih.gov/geo/query/acc.cgi?acc=GSE146672).All source data are provided in a Source Data file.

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

## Acknowledgements

We thank Michelle Itano in the UNC Neuroscience Microscopy Core for technical assistance with imaging experiments and Gabriela Salazar and Eric McCoy for technical assistance. This research was supported by the National Institute of Environmental Health Sciences of the National Institutes of Health (DP1ES024088, R56ES028236, R35ES028366, M.J.Z.), the National Institute of Aging (R56AG058663, M.J.Z. and M.J. M), the Owens Family Fund (M.J.M), and the microscopy core was supported by the Eunice Kennedy Shriver National Institute of Child Health and Human Development (U54HD079124) and NINDS (P30NS045892). J.M.S. was supported by The Eunice Kennedy Shriver National Institute of Child Health and Human Development (U54HD079124) and NINDS (P30NS045892).

## Author contributions

M.J.Z., G.F., and A.M.M. designed the study, G.F. performed all mouse experiments, single-molecule in situ hybridization, image quantification, and data analysis, B.T.-B. performed immunohistochemistry, J.K.N performed single-cell RNA-sequencing and analysis, H.M. provided assistance with experiments, H.Y. and Z.L. performed PET imaging and 18F-PBR111 synthesis, W.D.C. and M.J.M. performed CNV data acquisition, W.D.C., M.J.M., and J.M.S. performed CNV analysis, G.F. and M.J.Z. wrote the manuscript.

## Competing interests

The authors declare no competing interests.
