## [Peer Review File · Nature Communications]

Reviewers' Comments:

Reviewer #1:

Remarks to the Author:

Zylka et al generated a Top1 conditional knockout mouse in which they deleted Top1 in the cerebral cortex and hippocampus. The authors report specific vulnerability of the lower cortical neurons to Top1 loss which is associated with increased DNA damage, apoptosis, cell loss and juvenile mortality approx. 2 weeks after birth. The authors go on and show that cell death is at least in part mediated by PARP1 hyperactivation and NAD depletion since supplementation with an NAD precursor partially restored cell loss and improved life span. Overall, the manuscript is well-written and reports an exciting biology of Top1 in the cortex that will attract broad readership. It triggers many interesting questions, some of which need to be addressed before publication, as outlined below:

1. It is not clear from the outset why the authors decided to focus on excitatory neurons and not inhibitory GABAergic neurons. Do the latter neurons express shorter genes that are less likely to be dependent on Top1? What about cerebellar neurons?
2. The elevated microglia activation and astrogliosis suggest a widespread disruption of protein homeostasis. Is there evidence for protein aggregation or defective autophagy (e.g. p62 aggregation which is a marker of many neurodegenerative disorders associated with aberrant Top1 activity)?
3. The authors only rely on γ H2AX foci measurements as evidence of DNA damage, yet they don't show focus formation but rather a pan-nuclear staining. This data need strengthening with other DNA damage makers such as 53BP1 and more direct readouts such as comet assays in cultured neurons.
4. It is not clear how Top1 deletion in the cortex specifically impact lower cortical neurons. The authors provide interesting explanations that can readily be tested and will improve the manuscript. For example, comparing the expression of the long genes that are differentially expressed in L5 neurons in cTop1 and WT models.
5. Do lower cortical neurons accumulate more R-loops /DNA damage or is the threshold of DNA damage in these neurons lower than upper cortical neurons? Is gene length the only determining factor for the increased demand on Top1?
6. The aberrant accumulation of Top1-DNA breaks has been shown to associate with multiple ataxias and C9-ALS. Which is more toxic, not having Top1 or the aberrant accumulation of Top1-cc during transcription? Addressing this question experimentally is perhaps beyond the scope of this study but the reader will benefit from highlighting these important differences in the discussion.
7. NR supplement improved life span by 5 days and partially rescued neuronal death whereas deletion of one copy of p53 increased L5 cortical thickness without detectable effect on life span or neuron death. Is cell death in Top1 deficient neurons p53 independent? Will the combined effects of p53 partial deletion and NR supplementation not provide further improvement to life span? ...or is life span not dependent on cortical neuronal survival?

Reviewer #2:

Remarks to the Author:

The authors generate mice in which topoisomerase 1 is selectively deleted from neurons in the hippocampus and cerebral cortex. They show that the TOP1 deficient neurons exhibit progressive accumulation of DNA damage followed by PARP activation, cell degeneration and eventual death of the mice. Treatment of the mice with nicotinamide riboside and suppression of p53 ameliorated the neuronal degeneration and extended lifespan.

The findings are novel and of broad interest to the fields of neuroscience, DNA damage and cell death. The data appear solid and comprehensive. The manuscript is written well. I have no concerns.

Reviewer #3:

Remarks to the Author:

This study investigates the role of topoisomerase 1 (TOP1) in the development and maintenance of postmitotic neurons. To this end, the authors generated a conditional mouse line by crossing the Nex-Cre mice, which target recombination in excitatory neurons from E11.5, with TOP1-floxed mice. Their results showed that Nex-Cre;TOP1-fl/fl mice (cKO) exhibited a severe arrest in postnatal brain growth after postnatal day 7 (P7), accompanied by motor coordination phenotypes. At the cellular level, loss of TOP1 led to the developmental arrest of excitatory neurons in layers 5-6 from P2 to P7. In addition, excitatory neurons in all layers underwent progressive degeneration of from P7 to P12, characterized by elevated DNA damage, PAR activation, mitochondrial abnormalities, reduced NAD⁺ levels, astrogliosis and microgliosis. Interestingly, NAD⁺ supplementation using nicotinamide riboside (NR) partially rescued the cKO phenotypes, so as deleting the p53 gene.

Overall, this is an interesting paper that reports a very robust neurodevelopmental and neurodegenerative phenotypes in cKO mice that lack TOP1 in excitatory neurons. The identification of the impaired DNA damage repair pathway in TOP1 cKO is quite intriguing, albeit not entirely unexpected, given the essential role of TOP1 in DNA conformation and transcription. In addition, the characterizations of the excitatory neuron phenotypes in TOP1 cKO are quite thorough. Despite these merits, however, there are a few significant issues that need to be addressed.

1. The evidence supporting PAR activation in cKO mouse brain is very weak. The results in Figure 6a show a very faint smear of the signal for PAR protein in cKO brain. These results need to be repeated, validated and quantified to provide more convincing evidence. In addition, it will be important to demonstrate that PAR downstream targets are also activated in cKO brain.
2. Similar to point #1, the results of copy number variations (CNV) in cKO mouse brain are quite variable. Of the three cKO mice, only cKO #4902 show convincing CNVs in chromosome 6 (and maybe in chromosomes 8, 10 or 12). Given these marked variabilities, the contribution of CNV remains unclear.
3. The results from rescue experiments, using NR treatment and p53 deletion, appear to be very modest. For instance, although NR treatment seems to drastically reduce cleaved caspase 3+ and TUNEL+ cells in the cortex of cKO mice, the improvement of neuronal density (in Figure 6f) is very modest, though the images of NR-treated cKO brain seem to show a significant increase in neurons in L2-4, L5, and L6. It will be important to adapt a different quantification approach to characterize this phenotype.
4. Finally, the rescue results from p53 deleted cKO mice are quite puzzling (Figure 7c-d). The marked improvement in motor phenotypes is present in the absence of more drastic cellular rescue. It will be important to provide evidence regarding the DNA damage phenotype, including cleaved caspase 3 and TUNEL, in cKO;p53^{-/-} mice.

Response to reviews

We thank the Reviewers for their time and constructive comments. We spent the past ~six months conducting new experiments, including a small-scale single cell RNA-seq experiment, single molecule *in situ* hybridization and an additional single cell DNA-seq experiment. We feel these cutting-edge experiments and other changes thoroughly address each of the Reviewers major and minor comments. Our point-by-point responses are marked in blue. Additionally, we provided a marked-up version showing changes and additions relative to the initial submission.

Reviewer #1 (Remarks to the Author):

Zylka et al generated a Top1 conditional knockout mouse in which they deleted Top1 in the cerebral cortex and hippocampus. The authors report specific vulnerability of the lower cortical neurons to Top1 loss which is associated with increased DNA damage, apoptosis, cell loss and juvenile mortality approx. 2 weeks after birth. The authors go on and show that cell death is at least in part mediated by PARP1 hyperactivation and NAD depletion since supplementation with an NAD precursor partially restored cell loss and improved life span. Overall, the manuscript is well-written and reports an exciting biology of Top1 in the cortex that will attract broad readership. It triggers many interesting questions, some of which need to be addressed before publication, as outlined below:

1. It is not clear from the outset why the authors decided to focus on excitatory neurons and not inhibitory GABAergic neurons. Do the latter neurons express shorter genes that are less likely to be dependent on Top1? What about cerebellar neurons?

Evaluating how loss of *Top1* affected excitatory neurons was a massive undertaking. We were thus not able to evaluate loss of *Top1* in inhibitory neurons simultaneously. We initially focused on excitatory neurons because they far outnumber inhibitory neurons, making them easier to study, and since we previously found that long genes

are more strongly expressed in excitatory neurons relative to inhibitory neurons (Loo L. et al., 2019). We now more clearly delineate the rationale for focusing on excitatory neurons in our revised manuscript. And we added a sentence at the end of the Discussion, indicating how future studies could be focused on studying the effects of *Top1* deletion in inhibitory neurons, and other neurons.

We did not evaluate cerebellar neurons but instead, with new data, included analyses of hippocampal neurons. Hippocampal neurons, like cortical neurons, strongly express the *Neurod6* (NEX)-Cre allele (Goebbels S. et al., 2016) and degenerate following loss of *Top1*.

2. The elevated micoglia activation and astrogliosis suggest a widespread disruption of protein homeostasis. Is there evidence for protein aggregation or defective autophagy (e.g. p62 aggregation which is a marker of many neurodegenerative disorders associated with aberrant *Top1* activity)?

We performed immunostaining and Western blot analysis for p62 on P7 WT and cKO cortex of three mice per genotype but we did not find any difference in protein levels (see figure below). Therefore we don't have evidence for increased autophagy in *Top1* cKO mice.

3. The authors only rely on γ H2AX foci measurements as evidence of DNA damage, yet they don't show focus formation but rather a pan-nuclear staining. This data need strengthening with other DNA damage makers such as 53BP1 and more direct readouts such as comet assays in cultured neurons.

Yes, we detected γ H2AX foci and our quantifications were based on the number of γ H2AX foci/neurons. Neurons with γ H2AX foci>0 were considered γ H2AX positive. However we agree that the presentation of the data was not clear so we revised accordingly. We moved the high-magnification image in the previous Supplementary Figure 5a (showing γ H2AX foci formation in cKO mice) to a main figure (Figure 6b). Moreover we changed the definition of " γ H2AX positive cells" to "% of neurons with γ H2AX foci". As suggested by the Reviewer, we stained for Phospho-53BP1 and found an increase in signal in *Top1* cKO P7 cortical neurons (Supplementary Figure S6 b-c).

We believe that γ H2AX staining, Phospho-53BP1 staining and single-cell CNV analysis provide strong evidence to prove that *Top1* cKO mice exhibit increased DNA damage and subsequent genomic instability. Comet assays are a much less sensitive way of assessing DNA damage when compared to single cell genome sequencing, which detects precisely where DNA damage-induced mutations are located in each cell.

4. It is not clear how *Top1* deletion in the cortex specifically impact lower cortical neurons. The authors provide interesting explanations that can readily be tested and will improve the manuscript. For example, comparing the expression of the long genes that are differentially expressed in L5 neurons in c*Top1* and WT models.

We added new single-cell RNA-seq and single-molecule *in situ* hybridization data to answer this question (Figure 5). A comprehensive single-cell RNA-seq experiment is extremely costly and labor intensive, as we previously found (Loo L. et al., 2019). Thus, to limit cost, while still address this question, we performed a low cell number single-cell sequencing analysis of P7 WT and cKO cortical neurons. We found that long genes

were downregulated specifically in the excitatory neuron cluster--this was the only cell type that expresses *Neurod6* (NEX). We confirmed that *Top1* expression was reduced only in the excitatory neuron cluster, further confirming selectivity of our excitatory neuron knockout, except now using single-cell RNA-seq. We then performed single molecule *in situ* hybridization with probes to the longest gene that was downregulated in cKO excitatory neurons (*Ptprd*). We found that *Ptprd* was strongly reduced only in lower layer excitatory neurons of *Top1* cKO cortex, consistent with the hypothesis we raised in the discussion, now tested empirically. We added these new data to Figure 5, Supplementary Figure 5 and Supplementary Tables S1 and S2.

5. Do lower cortical neurons accumulate more R-loops /DNA damage or is the threshold of DNA damage in these neurons lower than upper cortical neurons? Is gene length the only determining factor for the increased demand on Top1?

Lower cortical neurons do accumulate more DNA damage, as shown in Figure 6 b-c and Supplementary Fig. S6a. We attempted to quantify R-loops in WT and cKO cortex by immunostaining with the S9.6 monoclonal antibody that recognizes DNA-RNA hybrids (Boguslawski S.J. et al., 1986). However, we only detected a signal in the nucleoli (see figure below). Nucleoli are known to accumulate R-loops due to the high rate of expression of ribosomal RNA genes (which are clustered and highly repetitive). Others likewise failed to identify R-loops in the brain of mice with a DNA-damage-induced form of neurodegeneration, using the same monoclonal S9.6 antibody (Yeo A.J. et al., 2014). Immunostaining with this antibody may thus lack the sensitivity needed to detect individual R-loops in non-repetitive regions of the genome, among other possible explanations. Sorting out the reason for why this antibody does/does not work well, or if R-loops are/are not formed is beyond the scope of our study. In the future, a more sensitive and quantitative way to detect R-loops genome-wide could be used (DRIP-seq). However, this technique would not allow us to distinguish different neuronal populations.

P7 cortex
DAPI Rloops (S9.6)

6. The aberrant accumulation of Top1-DNA breaks has been shown to associate with multiple ataxias and C9-ALS. Which is more toxic, not having Top1 or the aberrant accumulation of Top1-cc during transcription? Addressing this question experimentally is perhaps beyond the scope of this study but the reader will benefit from highlighting these important differences in the discussion.

This is a very interesting question. To address thoroughly we would need to generate and characterize an additional mouse line, for example TOP1 T718A. As we and others found, this mutation increases TOP1cc's (Mabb AM et al., 2016). However, as the Reviewer noted, adequately addressing this question is beyond the scope of the study.

TOP1cc's titrate out functional TOP1 via covalent attachment to DNA. In this way, TOP1cc's would be similar to loss of TOP1. However, the covalent attachment of TOP1 to DNA, if not resolved, would create a physical "roadblock" on the DNA and have a greater effect on gene expression than loss of TOP1. This is precisely what we found and discussed in our prior publication (Mabb AM et al., 2016).

7. NR supplement improved life span by 5 days and partially rescued neuronal death whereas deletion of one copy of p53 increased L5 cortical thickness without detectable effect on life span or neuron death. Is cell death in Top1 deficient neurons p53 independent? Will the combined effects of p53 partial deletion and NR supplementation not provide further improvement to life span? ...or is life span not dependent on cortical neuronal survival?

Is cell death in *Top1* deficient neurons *p53* independent?

Given the low probability (1/16-1/32 depending on the breeding scheme) of obtaining double *Top1* and *p53* cKO mice (*NexCre*^{+/-}::*Top1*^{fl/fl}::*p53*^{-/-}) and all the compound mutants required for the experiments, we only collected brains at P15, after performing the motor assays. At this time point, both TUNEL and cl-CASP3 were no longer detectable in the cortex of *Top1* cKO mice. We thus could not directly assess the effect of *p53* loss on cell death.

However, to address Reviewer 1, with new experiments, we quantified the numbers of CTIP2⁺ neurons (the neuronal population most vulnerable to *Top1* loss) in cortical layer 5 and 6 and in the CA1 hippocampal area of *Top1* cKO P15 mice lacking one or both copies of *p53* and we found a significant increase in these neurons compared with *Top1* cKO::*p53* WT mice (Fig. 8b-c). We also performed a staining for TOP1 and NEUN and found a clear increase in the survival of *Top1* cKO neurons upon deletion of *p53* (Supplementary Fig. S8). These data together with the increase in cortical thickness and hippocampal area, strongly suggest an involvement of *p53* in cell death induced by *Top1* loss (i.e. partial *p53* dependence).

Will the combined effects of *p53* partial deletion and NR supplementation not provide further improvement to life span? ...or is life span not dependent on cortical neuronal survival?

P53 acts downstream of PARP1 and other DNA-damage-induced pathways, so ablation of *p53* should mimic NR treatment by abrogating PARP1-dependent death and other death pathways induced by DNA damage. Consistent with this hypothesis, *p53* deletion led to a greater rescue of cell death in *Top1* cKO mice compared to NR treatment, yet did not rescue behavioral (motor) deficits, indicating that behavioral recovery is not dependent on neuron viability. NR likewise kept a greater proportion of neurons alive,

but did not cause any measureable behavioral recovery. We added more rigorous quantification of these data (see Fig. 7e-f; Fig 8b-c) and discussed the intriguing implication—that reducing neuronal loss (either by *p53* deletion or NR treatment) is not sufficient to limit behavioral decline when TOP1 function is disrupted.

We found no evidence that life span of *Top1* cKO mice was extended with partial or full deletion of *p53*, although the # of animals with the desired genotype that we evaluated was low (due to the low probability of obtaining the desired genotypes). Evaluating whether NR + *p53* deletion would have a greater effect on lifespan is an interesting question, but would require substantial amounts of time and effort. We feel our single cell/single molecule experiments provide greater (and more novel) insights into why neurons are vulnerable to loss of TOP1 than testing combined *p53* deletion w/ NR supplementation.

Reviewer #2 (Remarks to the Author):

The authors generate mice in which topoisomerase 1 is selectively deleted from neurons in the hippocampus and cerebral cortex. They show that the TOP1 deficient neurons exhibit progressive accumulation of DNA damage followed by PARP activation, cell degeneration and eventual death of the mice. Treatment of the mice with nicotinamide riboside and suppression of *p53* ameliorated the neuronal degeneration and extended lifespan.

The findings are novel and of broad interest to the fields of neuroscience, DNA damage and cell death. The data appear solid and comprehensive. The manuscript is written well. **I have no concerns.**

Reviewer #3 (Remarks to the Author):

This study investigates the role of topoisomerase 1 (TOP1) in the development and maintenance of postmitotic neurons. To this end, the authors generated a conditional mouse line by crossing the Nex-Cre mice, which target recombination in excitatory neurons from E11.5, with TOP1-floxed mice. Their results showed that Nex-Cre;TOP1-fl/fl mice (cKO) exhibited a severe arrest in postnatal brain growth after postnatal day 7 (P7), accompanied by motor coordination phenotypes. At the cellular level, loss of TOP1 led to the developmental arrest of excitatory neurons in layers 5-6 from P2 to P7. In addition, excitatory neurons in all layers underwent progressive degeneration of from P7 to P12, characterized by elevated DNA damage, PAR activation, mitochondrial abnormalities, reduced NAD⁺ levels, astrogliosis and microgliosis. Interestingly, NAD⁺ supplementation using nicotinamide riboside (NR) partially rescued the cKO phenotypes, so as deleting the p53 gene.

Overall, this is an interesting paper that reports a very robust neurodevelopmental and neurodegenerative phenotypes in cKO mice that lack TOP1 in excitatory neurons. The identification of the impaired DNA damage repair pathway in TOP1 cKO is quite intriguing, albeit not entirely unexpected, given the essential role of TOP1 in DNA conformation and transcription. In addition, the characterizations of the excitatory neuron phenotypes in TOP1 cKO are quite thorough. Despite these merits, however, there are a few significant issues that need to be addressed.

1. The evidence supporting PAR activation in cKO mouse brain is very weak. The results in Figure 6a show a very faint smear of the signal for PAR protein in cKO brain. These results need to be repeated, validated and quantified to provide more convincing evidence. In addition, it will be important to demonstrate that PAR downstream targets are also activated in cKO brain.

PARP1 is an enzyme that catalyzes the addition of Poly-(ADP)-ribose (PAR) to several proteins upon DNA damage, including PARP1 itself. In the Western blot that the Reviewer is referring to, we showed PAR (not PARP1). Since PAR is attached to many different proteins, anti-PAR antibody staining is expected to show a smear (Kam T.I. et

al., 2018; Hou Y. et al, 2018), as we found (Fig. 7a). Since the Reviewer was concerned about the quality of our Western blot, we performed this experiment again with a new set of P7 cortical lysates (from different mice) and obtained a clearer image (shown in Fig. 7a), which reproduced our previous results. We now also show quantified PARylated protein levels in Figure 7a instead of only reporting it in the Results section.

We agree with the Reviewer about the importance of showing PARP1 levels. Therefore, we performed Western blots for PARP1 (on 3 mice per genotype and replicated the results 3 times) and identified a significant increase in the enzyme protein levels, that further support the increase in PAR (Figure 7a).

As previously discussed, PARP1 has many downstream targets, many of which are not well characterized or can change based on the cell type and source of genotoxic stress (Jungmichel S. et al., 2013). We believe that identifying the PARP1 downstream targets in the *Top1* cKO excitatory neurons would be very interesting, but given the new data we present, we feel evaluating additional substrates (none of which have been rigorously confirmed in neurons) is beyond the scope of this study.

However, a well-characterized downstream effect of increased PAR levels is a decrease in NAD⁺ and subsequent cell death. We quantified NAD⁺ levels and found a deficit in P15 cKO animals (Table 1). Moreover, NAD⁺ supplementation improved both inflammation and apoptosis and led to an increase in CTIP2⁺ neurons (Fig. 7). We believe that these data provide strong evidence in support of an increased PARP1 activity in *Top1* cKO mice.

2. Similar to point #1, the results of copy number variations (CNV) in cKO mouse brain are quite variable. Of the three cKO mice, only cKO #4902 show convincing CNVs in chromosome 6 (and maybe in chromosomes 8, 10 or 12). Given these marked variabilities, the contribution of CNV remains unclear.

The CNV plots in our original submission were representative of single neurons, not mice (this is single-cell DNA-seq data). We believe our description of the results and the presentation of the data were not very clear and have modified them. We also repeated the single-cell DNA-seq experiment with more cells from a new pair of mice. This new experiment validated and extended the results of our first single-cell DNA-seq experiment.

Our aim was to test if the accumulation of DNA damage observed in cKO neurons resulted in an increased mutation burden. Therefore, in our original submission, we performed a small scale single-cell CNV analysis on 39 WT and 43 cKO neurons extracted from one WT and one cKO P7 mouse (previously shown in the main Figure, now shown in Supplementary Figure 6d). Instead of plotting data of only 3 cells per genotype, in the new version of the figure we show the overlap of the reads coming from all the WT (black line) and all the cKO neurons (red line). This plot is now in Supplementary Figure 6d. This first analysis showed an increase in the number of neurons with CNVs in *Top1* cKO cortex.

Given the small amount of cells analyzed and the limitation in the sensitivity of single-cell CNV techniques, we did not expect to find common mutation hotspots. This explains the variability in the CNV genomic locations and size pointed out by the Reviewer. To further validate the increased somatic mutation phenotype in the *Top1* cKO cortex, we repeated this experiment with more cells extracted from a new pair of mice and using a new, droplet-based whole genome amplification approach. These new single-cell DNA-seq data are shown in the Figure 6d and plotted as the overlap of the reads coming from all the WT (black line) and all the cKO neurons (red line). The list of CNVs identified is also provided in Supplementary Table S3. Again we observed increased numbers of neurons with CNVs (18.7% neurons in WT, 35.5% neurons in cKO), validating the presence of increased genomic instability in *Top1* cKO mice.

3. The results from rescue experiments, using NR treatment and p53 deletion, appear to be very modest. For instance, although NR treatment seems to drastically reduce cleaved caspase 3+ and TUNEL+ cells in the cortex of cKO mice, the improvement of neuronal density (in Figure 6f) is very modest, though the images of NR-treated cKO brain seem to show a significant increase in neurons in L2-4, L5, and L6. It will be important to adapt a different quantification approach to characterize this phenotype.

We agree that changes in cortical thickness alone do not fully represent changes in the cell death phenotype and that a more direct quantification of neuron survival is needed. We thus quantified the number of CTIP2⁺ neurons (the most affected neuronal subtype) in lower cortical layers and (with new data) hippocampus of the different treatment/genotype conditions (new data from hippocampus shown in Fig. 7e-f, Fig. 8a-b). We found a greater effect of NR treatment and p53 deletion on the number of CTIP2⁺ neurons when compared to cortical thickness or hippocampal area measures. The rescue in CTIP2⁺ neurons better correlates with the decreased apoptosis and neuroinflammation in NR-treated mice.

We also realized that showing only the cortical column of a specific area may not properly represent changes in the whole cortex therefore we decided to show images of the whole cortex and hippocampus (Fig. 7e, Fig 8b).

4. Finally, the rescue results from p53 deleted cKO mice are quite puzzling (Figure 7c-d). The marked improvement in motor phenotypes is present in the absence of more drastic cellular rescue. It will be important to provide evidence regarding the DNA damage phenotype, including cleaved caspase 3 and TUNEL, in cKO;p53^{-/-} mice.

Based on the Reviewer comment we realized that the presentation of our data was not very clear. Motor phenotypes were *not* rescued in *Top1* cKO mice lacking p53. These *Top1* cKO::p53 KO animals only showed a rescue in cortical thickness and hippocampal area measures. This was also supported by an almost complete rescue in the number

of CTIP2⁺ neurons in these areas. As we discussed in response to point #7 from Reviewer 1, given the low probability (1/16-1/32 depending on the breeding setting) of obtaining double *Top1* and *p53* cKO mice (*NexCre^{+/-}::Top1^{fl/fl}::p53^{-/-}*) and all the compound mutants required for the experiments, we only collected brains at P15, after performing the motor assays. Both TUNEL and cl-CASP3 peak at P7 in *Top1* cKO cortex and disappear by P15. Therefore we could not directly assess the effect of *p53* loss on cell death. However, with new data (Fig. 8b-c), we quantified the numbers of CTIP2⁺ neurons (the neuronal population most vulnerable to *Top1* loss) in cortical layer 5 and 6 and in the CA1 hippocampal area of *Top1* cKO P15 mice lacking one or both copies of *p53*. We found a significant increase in these neurons compared with *Top1* cKO::*p53* WT mice (Fig. 8b-c). We also performed a staining for TOP1 and NEUN and found a clear increase in the survival of *Top1* cKO neurons upon deletion of *p53* (Supplementary Fig. 8). These data together with the increase in cortical thickness and hippocampal area strongly suggest an involvement of *p53* in cell death induced by *Top1* loss.

Since *p53* acts downstream of DNA damage, loss of *p53* should not affect the amount of DNA damage accumulated by *Top1* cKO neurons, therefore we don't believe that staining for γ H2AX would be informative.

Lastly, since we added more rigorous quantification of these NR and *p53* data (see Fig. 7e-f; Fig 8b-c), we further discussed the intriguing implications of these data—that reducing neuronal loss (either by *p53* deletion or NR treatment) is not sufficient to limit behavioral decline when TOP1 function is disrupted.

Reviewers' Comments:

Reviewer #1:

Remarks to the Author:

I applaud the authors for their efforts, I have no further comments.

Reviewer #3:

Remarks to the Author:

The revised manuscript has fully addressed my comments/suggestions. I have no further concerns.

UNC
SCHOOL OF MEDICINE
Neuroscience Center

March 17, 2020

RE: NCOMMS-19-20613

Dear Reviewers,

We are glad that our efforts addressed all the Reviewers' comments. We thank the Reviewers for their time and constructive comments.

Sincerely,

Mark J. Zylka, Ph.D.

W.R. Kenan, Jr. Distinguished Professor

Director, UNC Neuroscience Center

REVIEWERS' COMMENTS:

Reviewer #1 (Remarks to the Author):

I applaud the authors for their efforts, **I have no further comments.**

Reviewer #3 (Remarks to the Author):

The revised manuscript has fully addressed my comments/suggestions. **I have no further concerns.**